# DeepHalo: A Neural Choice Model with Controllable Context Effects

**Shuhan Zhang**
The Chinese University of Hong Kong, Shenzhen
shuhanzhang@link.cuhk.edu.cn

**Zhi Wang**
University of Toronto
zhiss.wang@utoronto.ca

**Rui Gao**
The University of Texas at Austin
rui.gao@mccombs.utexas.edu

**Shuang Li**[*]
The Chinese University of Hong Kong, Shenzhen
lishuang@cuhk.edu.cn

## Abstract

Modeling human decision-making is central to applications such as recommendation, preference learning, and human-AI alignment. While many classic models assume context-independent choice behavior, a large body of behavioral research shows that preferences are often influenced by the composition of the choice set itself—a phenomenon known as the context effect or Halo effect. These effects can manifest as pairwise (first-order) or even higher-order interactions among the available alternatives. Recent models that attempt to capture such effects either focus on the featureless setting or, in the feature-based setting, rely on restrictive interaction structures or entangle interactions across all orders, which limits interpretability. In this work, we propose DeepHalo, a neural modeling framework that incorporates features while enabling explicit control over interaction order and principled interpretation of context effects. Our model enables systematic identification of interaction effects by order and serves as a universal approximator of context-dependent choice functions when specialized to a featureless setting. Experiments on synthetic and real-world datasets demonstrate strong predictive performance while providing greater transparency into the drivers of choice. **Code available at:** https://github.com/Asimov-Chuang/DeepHalo.

## 1 Introduction

Choice modeling [McFadden, 2001] provides a principled framework for capturing and predicting human preferences, making it central to human-in-the-loop systems. It enables personalized recommendations in online platforms [Resnick and Varian, 1997], preference inference in inverse reinforcement learning [Zeng et al., 2025], and reward alignment in large language models [Rafailov et al., 2023]. Across these applications, choice models offer a structured way to integrate human decision patterns into machine learning systems.

Traditional choice models often assume that each alternative's utility is independent of the others, implying stable preferences unaffected by context. However, human decisions frequently exhibit the context-dependent effect, also known as the *Halo effect* [Thorndike et al., 1920] in cognitive science, where the composition of the choice set influences preferences. For example, dominated options can increase the appeal of dominating ones (decoy effect [Huber et al., 1982]), extreme options can shift preference toward intermediates (compromise effect [Simonson and Tversky, 1992]), and similar alternatives can cannibalize each other's appeal (similarity effect [Tversky and Simonson, 1993]).

---

[*]Corresponding author.

39th Conference on Neural Information Processing Systems (NeurIPS 2025).

Such effects violate core assumptions of random utility models and call for more flexible approaches that capture interactions among alternatives.

While there have been choice models aiming to capture context effects, they often face structural limitations. Many rely on one-hot encodings and ignore rich feature information, and restrict interactions to first-order (pairwise) terms [Maragheh et al., 2018, Seshadri et al., 2019, Yousefi Maragheh et al., 2020b, Seshadri et al., 2020, Ko and Li, 2024]. On the other hand, Neural models offer flexibility and can incorporate features [Wong and Farooq, 2021, Wang et al., 2023], but their complex architectures hinder interpretability by entangling lower- and higher-order effects. As a result, existing methods often face a trade-off between expressiveness and transparency in modeling context-sensitive choice.

To bridge this gap, we propose a modeling framework that is both expressive enough to capture complex higher-order interactions among alternatives with features, and structured enough to systematically disentangle and interpret these effects. We begin by decomposing utility into interpretable, feature-based components—base utility, pairwise interactions, and higher-order effects—and characterize the associated permutation equivariance properties. This decomposition informs the design of a neural architecture with inductive biases aligned with the combinatorial structure of context effects. When specialized to the classic featureless setting (i.e., alternatives without access to their features), our model serves as a universal approximator of context-dependent choice functions. Additionally, we introduce a principled method for identifying interaction effects by order. We evaluate the model on hypothetical, synthetic, and real-world datasets—with and without alternative features—and show that it matches or exceeds the predictive performance of state-of-the-art baselines while providing greater interpretability and insight into the underlying drivers of choice behavior.

## 1.1 Literature Review

A substantial body of early work in marketing and cognitive science has examined context-dependent effects [Simonson, 1989, Tversky and Simonson, 1993, Simonson and Tversky, 1992, Tversky and Kahneman, 1991, Huber et al., 1982]. Rieskamp et al. [2006] provides a comprehensive review of empirical findings that challenge the assumptions of rational choice theory. The universal logit model—also known as the mother logit—was introduced by McFadden et al. [1977] and shown to approximate any discrete choice probability function [McFadden, 1984] by allowing for context-dependent utilities. Batsell and Polking [1985a] further specified this framework as a high-order interaction utility model. Besides, various context-dependent models have been proposed. These include models with menu-dependent consideration sets [Brady and Rehbeck, 2016], stochastic preference structures [Berbeglia and Venkataraman, 2018], pairwise utility formulations [Ragain and Ugander, 2016], contextual ranking models [Seshadri et al., 2020, Bower and Balzano, 2020, Makhijani and Ugander, 2019], and welfare-based frameworks that capture substitutability and complementarity [Feng et al., 2018].

This work takes [Batsell and Polking, 1985a] as a conceptual starting point. While theoretically expressive, this model presents significant estimation challenges, as the number of interaction terms can grow exponentially with the size of the choice set. To address tractability, much of the existing literature focuses on first-order context effects, which offer a compromise between interpretability and computational efficiency. These models are typically applied in featureless settings with small item universes, as their complexity still scales quadratically with the number of alternatives. The first-order interaction model—also known as the context logit or Halo model—has been explored in Maragheh et al. [2018], Seshadri et al. [2019], and Yousefi Maragheh et al. [2020b]. To capture low-rank structures, Li et al. [2021] and Ko and Li [2024] incorporate self-attention mechanisms into the Halo model. Separately, column generation approaches have been proposed for estimating generalized stochastic preference models [Berbeglia and Venkataraman, 2018], as in Jena et al. [2022]. Tomlinson and Benson [2021] extends the Halo model to incorporate features, but its reliance on a linear context structure limits its capacity to capture richer interactions in high-dimensional settings.

To address scalability and capture richer nonlinear context effects, recent work has explored deep learning architectures. Deep Sets [Zaheer et al., 2018], which ensure permutation invariance, have been adapted to choice modeling through set-dependent aggregation mechanisms [Rosenfeld et al., 2020], though primarily in featureless settings. Pfannschmidt et al. [2022] propose two neural architectures that incorporate features, one limited to first-order effects and the other entangling all higher-order interactions, making interpretation difficult. Residual net-based [Wong and Farooq, 2021] and transformer-based [Wang et al., 2023, Peng et al., 2024, Yang et al., 2025] models integrate

residual connection and/or attention mechanisms, improving predictive performance but often at the expense of interpretability, given that all-order interactions are entangled. To address this, we omit softmax normalization and further observe that the dot-product attention mechanism is not essential for modeling decomposable utility functions. This insight motivates a simplification of the architecture. While our model draws conceptual inspiration from Transformer-style aggregation, it diverges in architectural design to retain deep architectures' expressiveness while enabling explicit control over interaction order, yielding a transparent and interpretable framework for context-dependent choice.

Several related approaches address choice modeling from different angles. Context-independent models such as TasteNet [Han et al., 2020], RUMnet [Aouad and Désir, 2023], and NN-Mixed Logit [Wang et al., 2024] assume stable alternative utilities across choice sets. Graph-based models [Tomlinson and Benson, 2024, Zhang et al., 2024] capture alternative interactions using Graph Neural Network. Context-dependent models have also been explored in tasks like choice-set optimization [Tomlinson and Benson, 2020].

## 2 Preliminaries

### 2.1 Choice Model

Let $\mathcal{S}$ denote a finite universe of alternatives. A choice set is any non-empty subset $S \subseteq \mathcal{S}$, and a choice model specifies a probability distribution $\mathbb{P}_j(S)$ over the elements of $S$. That is, for any $j \in S$, $\mathbb{P}_j(S)$ denotes the probability that alternative $j$ is chosen from the set $S$.

A general and expressive framework for modeling such probabilities is McFadden's *universal logit model* [McFadden et al., 1977, McFadden, 1984]. For each $j \in \mathcal{S}$, define a utility function $u_j(S)$, which represents the utility of alternative $j$ when the offered set is $S$. Given a choice set $S$, the probability of choosing alternative $j \in S$ is defined as

$$\mathbb{P}_j(S) = \frac{\exp\left(u_j(S)\right)}{\sum_{k \in S} \exp\left(u_k(S)\right)}.$$

This formulation is called the universal logit model because it can approximate any choice function by defining the utility function $u_j(S)$ as the logarithm of the probability of choosing $j$ from the set $S$. It generalizes the Multinomial Logit model, which assumes that $u_j(S) = v_j$ is constant and independent of the choice set.

### 2.2 Context Effects

In general, the utility of an alternative $j \in \mathcal{S}$ can depend not only on its own characteristics but also on the specific composition of the choice set $S \subseteq \mathcal{S}$ in which it appears. That is, the presence or absence of other alternatives in $S$ may influence the perceived attractiveness of $j$. This phenomenon, known as the Halo effect or context effect [Tversky and Simonson, 1993], represents a departure from classical assumptions in discrete choice modeling, such as the *Independence of Irrelevant Alternatives*, which posit that utility is context-independent.

In the literature, context effect is typically discussed in the *featureless* setting, where each alternative has no features and is represented by a one-hot vector. To formally characterize context effects, we consider the following inclusion-exclusion-style decomposition of the utility function $u_j(S)$ [Batsell and Polking, 1985a, Seshadri et al., 2019]:

$$u_j(S) = v_j(\varnothing) + \sum_{j_1 \in S \setminus \{j\}} v_j(\{j_1\}) + \sum_{j_1 \neq j_2 \in S \setminus \{j\}} v_j(\{j_1, j_2\}) + \cdots + v_j(S \setminus \{j\}), \tag{1}$$

where the set function $v_j(\cdot)$ represents the marginal contribution of a subset $T \subseteq S \setminus \{j\}$ to the utility of alternative $j$. Here $T$ is a subset that exclude the item $j$ itself and thus emphasizes the context information. In this decomposition, $v_j(\varnothing)$ denotes the intrinsic or context-independent utility of $j$; $v_j(\{j_1\})$ captures first-order (pairwise) effect, such as how the presence of a single other alternative $j_1$ influences the utility of $j$; $v_j(\{j_1, j_2\})$ captures second-order effects, reflecting how pairs of alternatives jointly influence the utility of $j$; and so on up to $v_j(S \setminus \{j\})$, which encodes the full high-order effect of all remaining alternatives.

The decomposition (1) makes it possible to disentangle the utility impact of different interaction levels, allowing us to model subtle behavioral phenomena that cannot be captured by additive or

context-independent utility specifications. If all first-order and above terms vanish (i.e., $v_j(T) = 0$ for all $|T| \geq 1$), the model reduces to a context-independent utility framework. Many well-documented behavioral choice anomalies can be understood as manifestations of non-zero context effects. We follow the framework of Yousefi Maragheh et al. [2020b] to reinterpret common effects:

- *Decoy effect:* The presence of a clearly dominated alternative $k$ increases the utility of a dominating option $j$, reflected in $v_j(\{k\}) > 0$.

- *Similarity effect:* The utility of alternative $j$ decreases due to the presence of a similar option $k$, indicated by a negative first-order term $v_j(\{k\}) < 0$.

- *Compromise effect:* The presence of extreme options $k$ and $l$ enhances the attractiveness of an intermediate option $j$, modeled by a positive second-order interaction $v_j(\{k, l\}) > 0$.

## 3  Modeling the Context Effects in the Presence of Features

We now formalize our modeling framework of context effects in the settings where each alternative is associated with a feature vector. Let $\mathcal{S}$ be the universe of all alternatives, and suppose that each item $j \in \mathcal{S}$ is described by a feature vector $x_j \in \mathbb{R}^{d_x}$. Throughout, we assume the size of any choice set $S$ is upper bounded by $J$. To maintain a consistent domain for the utility function $u_j(\cdot)$, we assume the choice set $S$ is always padded (if necessary) to a fixed size $J$ using placeholder or null alternatives. Equivalently, when $|S| < J$, one may define $u_j = -\infty$ for $j = |S| + 1, \ldots, J$, implying that the choice probability is zero for these alternatives. The value of the utility function is taken from the extended real line $\bar{\mathbb{R}} = \mathbb{R} \cup \{-\infty\}$. In actual implementation, we apply a binary mask $\mu \in \{0, 1\}^J$ to ensure numerical stability. Our goal is to model the utility of each alternative $j \in S$ as a function of its feature vectors, in a way that accounts for context-dependent effects among alternatives.

### 3.1  Utility Decomposition and Permutation Equivariance

Observe that the summation in (1) is over unordered subsets $S$, thereby the utility function $u_j(S)$ is invariant to permutations of the other alternatives in $S$. Extending to the feature-based setting, this means the utility assigned to alternative $j$ depends only on the feature vectors of $S \setminus \{j\}$ and not on the order in which the alternatives appear. On the other hand, to facilitate neural network parameterization that will be discussed in the next subsection, we treat the choice set $S$ as an ordered tuple of indices $(1, \ldots, |S|)$, and represent the corresponding input feature matrix as

$$X_S := [x_1, \ldots, x_{|S|}, O_{J-|S|}] \in \mathbb{R}^{d \times J},$$

where $O_{J-|S|}$ is a $d \times (J - |S|)$ zero matrix to pad the feature matrix to a fixed size $d \times J$. This ordering enables the use of standard neural net architectures that operate over fixed-format inputs. However, since the true choice behavior is inherently invariant to the ordering of alternatives, it is critical that the model respects this symmetry. This motivates the explicit incorporation of such invariance into the model design.

To formalize this idea, with slight abuse of notation, we denote the utility function as $u_j(X_S)$ to emphasize that it is a function of the feature matrix $X_S$, which is ordered by the indices of the alternatives in $S$. We require that the utility function $u_j(X_S)$ satisfies the property of *permutation equivariance* with respect to the choice set $S$. Recall a function $f : \mathbb{R}^{d \times J} \to \bar{\mathbb{R}}^J$ is *permutation equivariant* [Zaheer et al., 2018] if

$$f_j(x_{\pi(1)}, \ldots, x_{\pi(J)}) = f_{\pi(j)}(x_1, \ldots, x_J), \quad \forall j = 1, \ldots, J,$$

for any permutation $\pi$ of the indices $1, \ldots, J$. This means that if the input feature vectors are permuted from $(x_1, \ldots, x_J)$ to $(x_{\pi(1)}, \ldots, x_{\pi(J)})$, the output of the function is permuted accordingly. We have the following result.

**Proposition 1.** *Every utility function* $u : \mathbb{R}^{d_x \times J} \to \bar{\mathbb{R}}^J$ *that is permutation equivariant can be decomposed as*

$$u_j(X_S) = \sum_{T \subset S \setminus \{j\}} v_j(X_{T \cup \{j\}}), \tag{2}$$

*where* $v_j$ *is a function over subsets of feature vectors that includes* $x_j$ *and is itself permutation equivariant in its arguments.*

Here, $X_{T \cup \{j\}} \in \mathbb{R}^{d_x \times J}$ denotes the matrix formed from the feature matrix $X_S$ by replacing the feature vectors not in the subset $T \cup \{j\}$ with zero. Proposition 1 shows that any permutation-equivariant utility function admits a representation as a sum over permutation-equivariant functions of subsets of feature vectors. This formulation mirrors the featureless decomposition in (1), but with utility now defined as functions of feature vectors. In particular, if each alternative is represented by a one-hot vector, then $v_j(\cdot)$ becomes a table over discrete subsets and (2) reduces to the original context-effect decomposition (1) without features. Intuitively, the function $v_j(X_{T \cup \{j\}})$ captures the contribution to utility from the interaction between $x_j$ and each subset $T$ of the remaining alternatives. In our constructive proof, we define $v_j$ as

$$v_j(X_{T \cup \{j\}}) := \sum_{R \subset T} (-1)^{|T|-|R|} u_j(X_{R \cup \{j\}}), \tag{3}$$

where the factor $(-1)^{|T|-|R|}$ refers to the inclusion–exclusion inversion coefficient on the Boolean lattice of subsets. This provides an explicit decomposition form for the First-Evaluate-Then-Aggregate (FETA) approach in Pfannschmidt et al. [2022], which only focuses on the first-order effect due to scalability limitations in modeling higher-order interactions; whereas their First-Aggregate-Then-Evaluate (FATE) architecture entangles all higher-order effects. In contrast, our explicit decomposition allows us to construct neural architectures that capture higher-order effects and systematically identify them, as detailed in the sections that follow.

## 3.2 Neural Net Parameterization

Based on the discussion above, we can now construct a neural network architecture, termed as DeepHalo, to parameterize the utility function $u_j(X_S)$ in a way that captures context effects of varying orders. Using (2), we rewrite the utility function as

$$u_j(X_S) = \sum_{p=0}^{|S|-1} \sum_{T \subset S \setminus \{j\}, |T|=p} v_j(X_{T \cup \{j\}}).$$

The inner sum is over all subsets $T$ of size $p$ from the set $S \setminus \{j\}$, capturing the contribution of $p$-th order interactions. The outer sum is over all possible orders of interactions, from 0 to $|S| - 1$. We now describe a neural architecture that parameterizes $u_j(X_S)$ as a sum of interaction terms of increasing order.

**Pairwise Interactions** We start with the first-order interaction:

$$u_j^{(1)}(X_S) := \sum_{T \subset S \setminus \{j\}, |T|=1} v_j(X_{T \cup \{j\}}) = \sum_{k \in S \setminus \{j\}} v_j(X_{\{j,k\}}).$$

One way to model the pairwise terms $\{v_j(X_{\{j,k\}})\}_{j \in S}$ is to use a multi-layer perceptron (MLP) that maps the concatenation $[x_j; x_k]$ to a vector in $\mathbb{R}^{|S|}$ [Pfannschmidt et al., 2022]. While expressive, this approach can be computationally expensive and does not scale well to higher-order interactions. To address this, we propose a more efficient form.

Let $z^0 := [\chi(x_j)]_{j \in S}$, where $\chi : \mathbb{R}^{d_x} \to \mathbb{R}^{d_0}$ is a shared nonlinear embedding function applied to each alternative. The use of a shared transformation ensures permutation equivariance. We define a context summary vector by linearly aggregating the embedded representations of all alternatives:

$$\bar{Z}^1 := \frac{1}{|S|} \sum_{k \in S} W^1 z_k^0 \in \mathbb{R}^H,$$

where $W^1 \in \mathbb{R}^{H \times d}$ is a shared linear projection across alternatives. Here, $H$ denotes the number of interaction heads, analogous to the number of channels or attention heads, which can be interpreted as $H$ different consumer types with different tastes. Thereby, $H$ controls the diversity of the interaction patterns. Let $\phi_h^1 : \mathbb{R}^d \to \mathbb{R}^d$, $h = 1, \ldots, H$, be nonlinear transformations applied to each embedded alternative. For each $j \in S$, define

$$z_j^1 := z_j^0 + \frac{1}{H} \sum_{h=1}^H \bar{Z}_h^1 \cdot \phi_h^1(z_j^0),$$

where the aggregated context $\bar{Z}_h^1$ is modulated by a nonlinear transformation of the base embedding $z_j^0$. This operation introduces first-order interactions, allowing the representation of alternative $j$ to depend on the presence of other alternatives in the set. The resulting utility $u_j^{(1)}(X_S) = \beta^\mathsf{T} z_j^1$ can be interpreted as the utility of alternative $j$ when considering the first-order interactions with all other alternatives in the choice set.

**Higher-order Interactions** We now extend the above structure to capture higher-order interactions. For each $l = 2, \ldots, L$, we recursively define

$$\bar{Z}^l := \frac{1}{|S|} \sum_{k \in S} W^l z_k^{l-1}, \tag{4}$$

$$z_j^l := z_j^{l-1} + \frac{1}{H} \sum_{h=1}^{H} \bar{Z}_h^l \cdot \phi_h^l(z_j^0), \tag{5}$$

where $W^l \in \mathbb{R}^{H \times d}$ and $\phi_h^l : \mathbb{R}^d \to \mathbb{R}^d$ are head-specific nonlinear transformations applied to the base embedding. This recursive formulation incrementally increases the order of interactions. The context summary $\bar{Z}^l$ aggregates global information from the previous layer, while the nonlinear transformations of the base embedding modulate how this summary influences each alternative. As a result, each layer introduces an additional higher-order interaction, while the residual connection preserves the lower-order effects accumulated from previous layers. Finally, we set

$$u_j(X_S) = \beta^\mathsf{T} z_j^L.$$

Permutation equivariance is maintained at each layer through symmetric aggregation in (4) [Zaheer et al., 2018]. The layered residual structure (5) builds the utility representation step by step, with the $l$-th layer capturing interaction effects up to order $l$. Compared to prior neural choice models Wong and Farooq [2021], Pfannschmidt et al. [2022], Wang et al. [2023], Peng et al. [2024], which entangle all interaction orders through deeply nested nonlinearities, our architecture incrementally and explicitly controls the interaction order at each layer. This yields both interpretability and architectural modularity, allowing practitioners to tailor the model's complexity to the degree of the context effects.

# 4 Discussions

In this section, we present alternative architectures for DeepHalo, establish its universal approximation property in the classic featureless setting, and discuss the identification of context effects; additional details are provided in the supplemental material.

## 4.1 Residual Connection for Large Choice Sets

The recursive structure defined in (4) uses exactly $l$ layers to capture $l$-th order interactions. When the choice set $S$ contains a large number of alternatives, the maximum interaction order $|S| - 1$ may be large, leading to very deep architectures. To improve computational efficiency while retaining expressiveness, we can consider a polynomial aggregation in place of (4):

$$\bar{Z}^l = \frac{1}{|S|} \sum_{k \in S} W^l \sigma(z_k^{l-1}), \tag{6}$$

where $\sigma$ is an element-wise polynomial activation, and consider a more flexible residual function:

$$z_j^l = z_j^{l-1} + \frac{1}{H} \sum_{h=1}^{H} \Phi_h^l\big(\bar{Z}_h^l, \phi_h^l(z_j^0)\big), \tag{7}$$

where $\Phi_h^l$ is a polynomial function on $\mathbb{R} \times \mathbb{R}$. For example, setting $\sigma$ to be quadratic and $\Phi_h^l$ to be a linear transformation of the second argument, we get a ResNet-like structure

$$z_j^l = z_j^{l-1} + \frac{1}{S} \sum_{k=1}^{S} W^l \sigma(z_k^{l-1}). \tag{8}$$

## 4.2 Specialization to the Featureless Setting

A special case of our framework arises in the featureless setting, where each alternative is identified solely by a unique index in the finite universe $\mathcal{S} = \{1, \ldots, J\}$. In this case, the feature vector of the $j$-th alternative, denoted by $\mathbf{1}_j^S$, is a one-hot vector of dimension $J$ with a one in the $j$-th position if $j \in S$ and zero elsewhere. Denote by $e_j$ the $j$-th unit vector in $\mathbb{R}^J$, and by $e_S = \sum_{j \in S} e_j$ be the indicator vector of the set $S$. Identifying $e_S$ with the set $S$, the utility decomposition (1) can be viewed as a degree-$(J-1)$ multivariate polynomial $u_j(e_S) = \sum_{p=0}^{J-1} w_p(e_S)$, where $w_p(e_S)$ is a degree-$p$ polynomial of $e_S$. We now show how our recursive architecture naturally expresses such polynomials.

Suppose the base embedding function $\chi$ is the identity map and let the interaction modulator $\phi_h^l(\mathbf{1}_j^S) = q_{hj}^l e_j$ if $j \in S$ and zero otherwise, where $q^l := [q_{hj}^l]_{hj} \in \mathbb{R}^{H \times J}$ is a learnable parameter matrix. Under this setup, the recursion (4)(5) becomes

$$\begin{cases} z_j^1 = z_j^0 + \frac{1}{HS} \sum_{h=1}^{H} \left( \sum_{k \in S} W_{hk}^1 \right) \cdot q_{hj}^1 e_j \\ z_j^l = z_j^{l-1} + \frac{1}{HS} \sum_{h=1}^{H} \left( \sum_{k \in S} W_{h,\cdot}^l z_k^{l-1} \right) \cdot q_{hj}^l e_j, \quad l = 2, \ldots, L, \end{cases} \tag{9}$$

where $W_{h,\cdot}^l \in \mathbb{R}^{1 \times J}$ denotes the $h$-th row of the matrix $W^l \in \mathbb{R}^{H \times J}$. Define a matrix $\Theta^l := \frac{1}{HS} \sum_{h=1}^{H} (q_{h,\cdot}^l)^\mathsf{T} W_{h,\cdot}^l \in \mathbb{R}^{J \times J}$, which has rank at most $H$. Let $y^l := \sum_{j=1}^{J} z_j^l \in \mathbb{R}^J$ denote the aggregate representation at layer $l$. Then, with $y^0 = e_S$, the recursion (9) implies

$$y^l = y^{l-1} + \Theta^l(y^{l-1} \odot e_S), \quad l = 1, \ldots, L,$$

where $\odot$ denotes the Hadamard product, and the utility of alternative $j$ is given by $u_j(S) = y_j^L$.

Thus, our model reduces to an $L$-layer residual network with residual connections between the input layer and each subsequent layer, where each layer applies a rank-$H$ linear transformation to a masked (context-aware) version of the previous layer's output. By induction, each $y^l$ is an $(l+1)$-degree polynomial of $e_S$. When $L = 1$, this model reduces to the lower-rank context-dependent random utility model of Seshadri et al. [2019] of the low-rank Halo MNL model [Ko and Li, 2024]. When $L = 1$ and $H = J$, it recovers the full-rank context-dependent random utility model of [Seshadri et al., 2019] or the contextual MNL (CMNL) model of Yousefi Maragheh et al. [2020b].

More generally, following Section 4.1, if we use an element-wise quadratic activation $\sigma(\cdot) = (\cdot)^2$, the recursion becomes

$$y^l = y^{l-1} + \Theta^l \sigma(y^{l-1}), \quad l = 1, \ldots, L. \tag{10}$$

Under this formulation, it takes at most $\lceil \log_2(J-2) \rceil$ layers to capture all orders of interactions of the $J$ alternatives.

## 4.3 Estimating the Context Effects

Observe that due to the translation invariance of the softmax function, the context effect coefficients $v_j(T), T \subset \mathcal{S} \setminus \{j\}$, in (1) are not directly identifiable from the choice probabilities unless additional linear constraints are imposed on these coefficients [Seshadri et al., 2019]. Nonetheless, the following quantity, which we call the *relative context effect*, is identifiable (see also Park and Hahn [1998a]):

$$\alpha_{jk}(T) = \big(v_j(T) + v_j(T \cup \{k\})\big) - \big(v_k(T) + v_k(T \cup \{j\})\big), \tag{11}$$

where $j, k \in \mathcal{S} \setminus T$. It captures the marginal effect of the subset $T$ on the utility difference between alternatives $j$ and $k$. To estimate $\alpha_{jk}(T)$, it suffices to first evaluate our neural network on subsets $R \cup \{j, k\}$ for all $R \subset T$, and then use the formula (3) to compute the four terms in (11) to obtain the desired quantity.

## 5 Empirical Study

In this section, we conduct experiments to evaluate the proposed model. Detailed experimental settings are provided in the supplementary material.

## 5.1 Featureless Datasets

**Hypothetical Data with Halo Effects** To demonstrate our model's efficacy in capturing and recovering context effects, we use a hypothetical beverage market share dataset, originally employed by Batsell and Polking [1985a] and Park and Hahn [1998a]. Suppose we have four products—*Pepsi*, *Coke*, *7-Up*, and *Sprite*—in the universe, denoted as $\mathcal{S} = \{1, 2, 3, 4\}$, respectively. Assume the market shares for all possible choice sets are as listed in Table 1a, and follow the hypothetical behavioral rules: (a) consumers view the products primarily as a choice between a cola and a non-cola; (b) they almost always choose *Pepsi* over *Coke* when both are available and a cola is desired; and (c) they almost always choose *7-Up* over *Sprite* when both are available and a non-cola is desired.

| Choice Set | 1 | 2 | 3 | 4 |
|:---:|:---:|:---:|:---:|:---:|
| (1,2) | 0.98 | 0.02 | – | – |
| (1,3) | 0.50 | – | 0.50 | – |
| (1,4) | 0.50 | – | – | 0.50 |
| (2,3) | – | 0.50 | 0.50 | – |
| (2,4) | – | 0.50 | – | 0.50 |
| (3,4) | – | – | 0.90 | 0.10 |
| (1,2,3) | 0.49 | 0.01 | 0.50 | – |
| (1,2,4) | 0.49 | 0.01 | – | 0.50 |
| (1,3,4) | 0.50 | – | 0.45 | 0.05 |
| (2,3,4) | – | 0.50 | 0.45 | 0.05 |
| (1,2,3,4) | 0.49 | 0.01 | 0.45 | 0.05 |

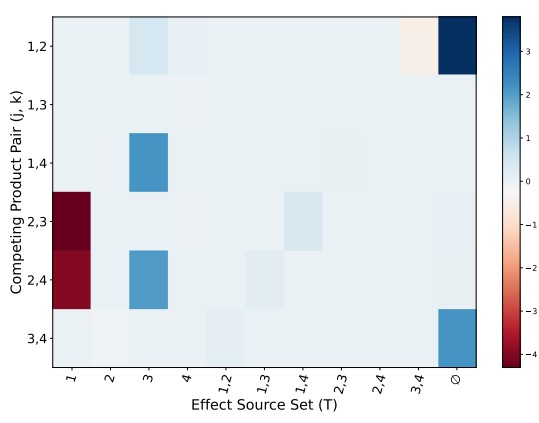

(a) Beverage market share          (b) Relative Halo effect

Figure 1: Market-share table (left) and relative Halo effect visualization (right)

For each conceivable choice set, we generate 2000 samples, where each choice outcome is drawn according to the market share as the choice probability. We then fit a quadratic-activated DeepHalo model with depth $L = 2$, as specified in Section 4.2. Subsequently, we recover the identifiable relative context effects following the procedure in Section 4.3 and visualize them as a heatmap in Figure 1b. Each colored cell represents the marginal influence of a source set $T$ on the relative utility between a product pair $(j, k)$. The column labeled by the empty set $(\varnothing)$ corresponds to the zero-order relative Halo effect between alternatives $j$ and $k$.

This visualization highlights the interpretability of our model. For example, in the first column, the red cells for pairs $(2, 3)$ and $(2, 4)$ indicate that, the presence of product 1 (*Pepsi*) lowers the utility of product 2 (*Coke*) relative to products 3 and 4 (*7-Up* and *Sprite*), making *Coke* less likely to be chosen. This exactly reflects the "Pepsi-over-Coke" preference captured by logic rule (b) above. A similar pattern appears in the top-right corner of the heatmap, where product 1 consistently has higher zero-order utility than product 2, reaffirming that *Pepsi* is inherently more preferred than *Coke*, regardless of context.

**Synthetic Data with High-order Effects** To empirically assess how model depth influences expressiveness under a fixed parameter budget, we construct a synthetic dataset with a universe of alternatives $\mathcal{S} = \{1, \ldots, 20\}$ and choice sets of fixed cardinality 15. For each such choice set, we sample a choice probability vector uniformly from the probability simplex on $\mathbb{R}^{15}$ and generate 80 i.i.d. choice observations. This results in a dataset with 1,240,320 training samples and can implicitly incorporate higher-order interactions up to 14-th order. We report the training root mean squared error (RMSE) of the predicted choice probabilities in Figure 2.

Recall from Section 4.2 that with quadratic activations, an $L$-layer network can represent interactions up to $2^{L-1}$-th order. In Figure 2a, we observe that when the number of parameters is fixed (either 200k or 500k), RMSE decreases significantly as depth increases up to 5, beyond which $2^{5-1}$ exceeds the choice set size of 15. For deeper models ($L \geq 6$), further performance gains are marginal. At fixed depth, models with more parameters perform slightly better, as expected, but increasing width alone does not compensate for insufficient depth. These results confirm that model expressiveness scales

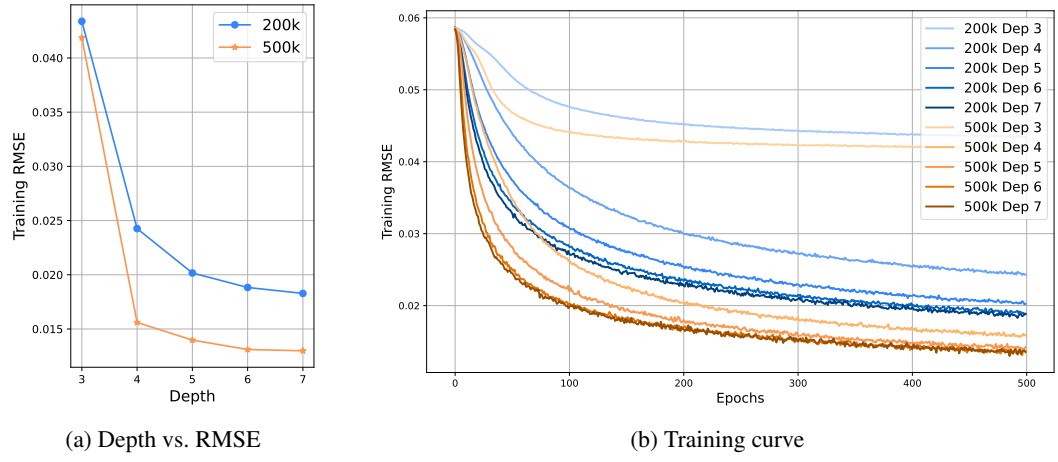

(a) Depth vs. RMSE               (b) Training curve

Figure 2: Effect of model depth on approximation error.

exponentially with depth and that depth is the dominant factor for capturing high-order interactions in choice behavior.

**Real Data**   To investigate the necessity of modeling high-order context effects, we evaluate the empirical performance of our featureless model on three real-world datasets: Hotel [Bodea et al., 2009], SFOwork and SFOshop [Seshadri et al., 2019]. The Hotel dataset records bookings from five continental hotels. Following the preprocessing steps in Berbeglia et al. [2022], we focus on the second hotel, where each assortment includes up to 11 alternatives (including a "leave" option). We use 1,845 observations for training and 465 for testing; due to the limited sample size, no separate validation split is held out. The SFOwork and SFOshop datasets contain travel mode choices in the San Francisco Bay Area, with the former focused on work trips and the latter on shopping trips. SFOwork provides 5,029 observations with up to 6 alternatives per choice set, and SFOshop provides 3,157 observations with up to 8 alternatives. Both datasets are partitioned into training, validation, and test sets in an 8:1:1 ratio. All models are trained and evaluated using the negative log-likelihood (NLL), and results are reported in Table 1, alongside baseline methods including Multinomial Logit (MNL), Multilayer Perceptron (MLP), and Contextual MNL (CMNL) [Yousefi Maragheh et al., 2020b]. Across all three datasets, DeepHalo achieves the lowest NLL among the compared models. This consistent performance highlights the value of modeling deep Halo effects: MNL and MLP do not account for context effects, and CMNL captures only first-order interactions.

Table 1: Negative log-likelihood (NLL) on the Hotel, SFOshop, and SFOwork datasets

| Model | Hotel | | SFOshop | | SFOwork | |
|---|---|---|---|---|---|---|
| | Train | Test | Train | Test | Train | Test |
| MNL | 0.7743 | 0.7743 | 1.7281 | 1.7262 | 0.9423 | 0.9482 |
| MLP | 0.7569 | 0.7523 | 1.5556 | 1.5523 | 0.8074 | 0.8120 |
| CMNL | 0.7566 | 0.7561 | 1.5676 | 1.5686 | 0.8116 | 0.8164 |
| DeepHalo (Ours) | **0.7479** | **0.7483** | **1.5385** | **1.5263** | **0.8040** | **0.8066** |

## 5.2 Interpretable Low-Order Modeling vs. Post Hoc Truncation

To assess the effect of explicit order control, we compare two models on a controlled synthetic dataset and the real SFOshop dataset: (1) DeepHalo($k$=2), constrained to second-order interactions, and (2) MLP+Trunc($k$), a parameter-matched MLP whose interaction terms are extracted and truncated to order $k$ at evaluation. Model expressiveness is measured by in-sample negative log-likelihood (NLL).

DeepHalo achieves consistently lower NLL by directly constraining interaction order, while MLP+Trunc matches its performance only when all higher-order terms are included. Increasing $k$ beyond the true interaction order can even worsen fit, as higher-order effects co-adapt with lower-order ones and their removal distorts predictions. On SFOshop, the NLLs of DeepHalo($k$=2) and

Table 2: In-sample NLL (lower is better) for MLP with truncation to order $k$ and DeepHalo constrained to second order.

| Dataset | Truncation order $k$ | MLP+Trunc($k$) NLL | DeepHalo($k=2$) NLL |
|---|---|---|---|
| Synthetic (true order $\leq 2$) | 2 | 0.7643 | 0.0922 |
| | 3 | 1.6429 | |
| | 4 | 0.8172 | |
| | 5 | 0.6719 | |
| | 6 | 0.3280 | |
| | 7 | 0.1502 | |
| | 8 | 0.0958 | |
| | 9 | 0.0910 | |
| SFOshop (human data) | 2 | 1.6065 | 1.5339 |
| | 3 | 1.6168 | |
| | 4 | 1.6834 | |
| | 5 | 1.5806 | |
| | 6 | 1.5406 | |
| | 7 | 1.5340 | |

full-order DeepHalo are nearly identical (1.5339 vs. 1.5331), revealing that in real-world scenarios, human choice behavior can be captured by low-order interactions.

## 5.3 Datasets with Features

We experiment on two real-world datasets of different scales: the smaller LPMC transportation dataset [Hillel et al., 2018] with 81,086 observations and the larger Expedia Hotel Choice dataset [Adam et al., 2013] with 275,609 transactions. The LPMC dataset, based on the London Travel Demand Survey, captures travel mode choices among walking, cycling, public transport, and driving, with each alternative described by 8 item-specific features and 17 shared features (e.g., cost, travel time, gender, and age). The Expedia dataset contains hotel search and booking records. After preprocessing, each choice set includes up to 38 hotels, with 35 item-specific and 56 shared features (e.g., price, star rating, destination).

Table 3: Negative log–likelihood (NLL) on the LPMC and Expedia datasets.

| Dataset | Split | MNL | MLP | TasteNet | RUMnet | DLCL | ResLogit | FateNet | TCNet | DeepHalo (Ours) |
|---|---|---|---|---|---|---|---|---|---|---|
| LPMC | Train | 0.8816 | 0.7062 | 0.7240 | 0.7041 | 0.7137 | 0.7032 | 0.6936 | 0.6645 | **0.6444** |
| | Test | 0.8643 | 0.6857 | 0.7001 | 0.6830 | 0.6977 | 0.6833 | 0.6774 | 0.6487 | **0.6430** |
| Expedia | Train | 2.6272 | 2.5637 | 2.5529 | 2.5440 | 2.5606 | 2.5462 | 2.5419 | **2.4865** | 2.5059 |
| | Test | 2.6244 | 2.5758 | 2.5776 | 2.5776 | 2.5601 | 2.5715 | 2.5515 | 2.5340 | **2.5263** |

We compare our model against a range of representative feature-based baselines, including classic models such as MNL and MLP, and recent neural approaches like TasteNet [Han et al., 2020], RUMnet[Aouad and Désir, 2023], ResLogit [Wong and Farooq, 2021], DLCL [Tomlinson and Benson, 2021], FateNet [Pfannschmidt et al., 2022], and TCNet [Wang et al., 2023]. Expressive models that account for context effects, including DeepHalo, FateNet, and TCNet, consistently outperform context-independent approaches (MNL, MLP, TasteNet, and RUMnet) and the remaining less expressive context-dependent ones on both datasets, highlighting the importance of modeling contextual interactions with sufficient flexibility in choice behavior. Among them, DeepHalo generalizes the best consistently while offering favorably improved interpretability compared to other deep neural network models.

## 6 Conclusion

We proposed a neural framework for context-dependent choice modeling that enables controlled modeling of interaction effects by order. While we focused on one specific parameterization of higher-order effects, the broader framework admits many alternatives. Exploring such variants offers a promising direction for developing interpretable high-capacity models with downstream applications in building AI systems that reason about and adapt to human behavior.

## Acknowledgments and Disclosure of Funding

This work was in part supported by the Key Program of the NSFC under grant No. 72495131, NSFC under grant No. 62206236, Shenzhen Stability Science Program 2023, Shenzhen Science and Technology Program ZDSYS20230626091302006, and Longgang District Key Laboratory of Intelligent Digital Economy Security.

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

# Appendix

## Appendix Contents

# A Proofs and Derivations

## A.1 Proof of Proposition 1

*Proof.* Consider the following form of $v$:

$$v_j(X_{T \cup \{j\}}) := \sum_{R \subset T} (-1)^{|T|-|R|} u_j(X_{R \cup \{j\}}). \tag{12}$$

Thanks to the permutation equivariance of $u_j$, the function $v_j$ given by (3) is permutation equivariant. Indeed, for every permutation $\pi$ on $T \cup \{j\}$, it holds that

$$v_j(X_{\pi(T \cup \{j\})}) = \sum_{Q \subset \pi(T)} (-1)^{|\pi(T)|-|Q|} u_j(X_{Q \cup \pi(j)})$$

$$= \sum_{R \subset T} (-1)^{|T|-|R|} u_{\pi(j)}(X_{R \cup \{j\}})$$

$$= v_{\pi(j)}(X_{T \cup \{j\}}),$$

where the second equality follows from the change of variable $R = \pi^{-1}(Q)$.

Next, let us verify that the function $v$ given in (3) can express the utility function $u_j(X_S)$. Indeed, for every $S$ and $j = 1, \ldots, |S|$, using (3) we write $\sum_{T \subset S \setminus \{j\}} v(x_j, X_T)$ as

$$\sum_{T \subset S \setminus \{j\}} \sum_{R \subset T} (-1)^{|T|-|R|} u(X_{R \cup \{j\}}).$$

By switching the order of summation, this equals

$$\sum_{R \subset S \setminus \{j\}} u(X_{R \cup \{j\}}) \sum_{T \supset R, T \subset S \setminus \{j\}} (-1)^{|T|-|R|}.$$

When $R = S \setminus \{j\}$, the inner sum is 1. Otherwise, the inner sum equals

$$\sum_{k=0}^{|S|-1-|R|} (-1)^k \binom{|S|-1-|R|}{k} = 0^{|S|-1-|R|} = 0.$$

This shows that

$$\sum_{T \subset S \setminus \{j\}} v(x_j, X_T) = u_j(X_S),$$

which completes the proof.

$\square$

## A.2 Derivation in Section 4.2

### A.2.1 Recursion for $y^l$

Let us first write $y^l = \sum_{j=1}^{J} z_j^l$ with $y^0 = e_S$ where $z^l$ is defined in (9). Since $y^0 = e_S = \sum_{j=1}^{J} e_j = \sum_{j=1}^{J} z_j^0$, it implies $\chi(x) = x$ being an identity mapping such that $z_j^0 = e_j$. Recall

$$\Theta^l = \frac{1}{HS} \sum_{h=1}^{H} (q_{h,\cdot}^l)^\mathsf{T} W_{h,\cdot}^l, \quad \Theta_{jk}^l = \frac{1}{HS} \sum_{h=1}^{H} q_{hj}^l W_{hk}^l.$$

We write

$$z_j^1 = e_j + \frac{1}{HS} \sum_{h=1}^{H} \left( \sum_{k \in S} W_{hk}^1 \right) \cdot q_{hj}^1 e_j = e_j + \sum_{k \in S} \Theta_{jk}^1 e_j.$$

It follows that

$$y^1 = \sum_{j=1}^{J} e_j + \sum_{j=1}^{J} \sum_{k \in S} \Theta_{jk}^1 e_j = \sum_{j=1}^{J} e_j + \sum_{j=1}^{J} \sum_{k \in S} \Theta_{jk}^1 e_k e_j = y^0 + \Theta^1(e_S \odot e_S) = y^0 + \Theta^1(y^0 \odot e_S).$$

For $l = 2, \ldots, L$,

$$y^l = \sum_{j=1}^J z_j^{l-1} + \sum_{j=1}^J \frac{1}{HS} \sum_{h=1}^H \left( \sum_{k \in S} W_{h,\cdot}^l z_k^{l-1} \right) \cdot q_{hj}^l e_j$$

$$= y^{l-1} + \sum_{j=1}^J \frac{1}{HS} \sum_{h=1}^H \sum_{k=1}^J W_{hk}^l q_{hj}^l y_k^{l-1} e_j$$

$$= y^{l-1} + \sum_{j=1}^J \sum_{k=1}^J \Theta_{jk}^l y_k^{l-1} e_j$$

$$= y^{l-1} + \Theta^l (y^{l-1} \odot e_S)$$

as desired.

### A.2.2 Universal approximation

We consider a flexible model by setting $y^l \in \mathbb{R}^{J'}$ with $J' > J$. Let $\Theta^1 \in \mathbb{R}^{J' \times J}$, $\Theta^l \in \mathbb{R}^{J' \times J'}$ for $l = 2, \ldots, L$ and $W^l \in \mathbb{R}^{J \times J'}$.

$$\begin{cases} y^1 = \Theta^1 e_S, \\ y^l = y^{l-1} + \Theta^l \left( y^{l-1} \odot y^1 \right), & l = 2, \ldots, L. \end{cases}$$

We show inductively that there exists $W^l \in \mathbb{R}^{J \times J'}$ such that $W^l y^l$ can represent any up to $l$-th order interactions.

We first consider the base case $l = 1$. By definition of $y^1$,

$$y_j^1 = \sum_{k=1}^J \Theta_{jk}^1 \mathbb{I}(k \in S).$$

For $j \in S$, we can further write

$$y_j^1 = \Theta_{jj}^1 \mathbb{I}(j \in S) + \sum_{k=1:k \neq j}^J \Theta_{jk}^1 \mathbb{I}(k \in S)$$

$$= v_j^1(\varnothing) \mathbb{I}(j \in S) + \sum_{k=1:k \neq j}^J v_j^1(\{k\}) \mathbb{I}(k \in S),$$

where $v_j^1(\varnothing) =: \Theta_{jj}^1$, $v_j^1(\{k\}) := \Theta_{jk}^1$. For $j > J$, it can be set to any function value parameterized by $\Theta_{j,\cdot}^1 e_S$ provided that $S$ is not empty, which is independent of other $y_j^{l-1}$'s with $j \leq J$. Without loss of generality, we can simply set $W^1 = [I_J; 0] \in \mathbb{R}^{J \times J'}$ to output $W^1 y^1 = \{y_j^1\}_{j \in S}$, which represent any first-order interaction model.

Now suppose $W^{l-1} y_j^{l-1}$ can represent any interaction model up to $(l-1)$-th order with $W^{l-1} = [I_J; 0] \in \mathbb{R}^{J \times J'}$, which means for any $j = 1, \ldots, J$,

$$y_j^{l-1} = \sum_{T \subseteq \{1, \ldots, J\}, |T| \leq l-1} v_j^{l-1}(T) \prod_{k \in T} \mathbb{I}(k \in S).$$

and for any $j > J$, $y_j^{l-1}$ can be set to any function value independent of other $y_j^{l-1}$'s with $j \leq J$.

For $j = 1, \ldots, J'$,

$$y_j^l = y_j^{l-1} + \sum_{j'=1}^{J'} \Theta_{jj'}^l y_{j'}^{l-1} \sum_{k=1}^J \Theta_{j'k}^1 \mathbb{I}(k \in S)$$

$$= y_j^{l-1} + \sum_{j'=1}^{J'} \Theta_{jj'}^l \sum_{T \subseteq \{1, \ldots, J\}, |T| \leq l-1} v_j^{l-1}(T) \prod_{k \in T} \mathbb{I}(k \in S) \sum_{k'=1}^J \Theta_{j'k'}^1 \mathbb{I}(k' \in S)$$

$$= \bar{y}_j^{l-1} + \sum_{|T| \subseteq \{1, \ldots, J\}, k' \in \{1, \ldots, J\}, |T| \cup \{k\} = l} \sum_{j'=1}^{J'} \Theta_{jj'}^l \Theta_{j'k'}^1 v_j^{l-1}(T) \prod_{k \in T \cup \{k'\}} \mathbb{I}(k' \in S),$$

where $\bar{y}_j^{l-1}$ is another function that can capture any up to $l-1$-th order interaction by combining original $y_j^{l-1}$ with the terms in the summation where $k \in T$. Since $\Theta_{jj'}^l$ is independent of $v_j^{l-1}(T)$, we can identify $\hat{v}_j^l(T \cup \{k\}) = \sum_{j'=1}^{J'} \Theta_{jj'}^l \Theta_{j'k'}^1 v_j^{l-1}(T)$ for any $T \cup \{k\} = l$, which use $J'$ basis to approximate any $l$-th order interaction. Thus

$$y_j^l = \bar{y}_j^{l-1} + \sum_{T \subseteq \{1,\dots,J\}, |T|=l} \hat{v}_j^l(T) \prod_{k \in T} \mathbb{I}(k \in S).$$

With $W^l = [I_J; 0] \in \mathbb{R}^{J \times J'}$, $W^l y^l = [y_j^l]_{j \in S}$ approximate any up to $l$-th order interactions. By induction, $y_j^L$ can capture up to $L$-th order interaction with sufficiently large $J'$, where $W^L$ can be set to $[I_J; 0] \in \mathbb{R}^{J \times J'}$ without loss of generality.

### A.2.3 Quadratic activation

Here we explain why the formulation (10) needs at most $\lceil 1 + \log_2(J-1) \rceil$ layers to capture all orders of interactions of the $J$ alternatives.

Starting $l = 1$, given $y^0 = e_S$,

$$y^1 = e_S + \Theta^1(e_S \odot e_S) = e_S + \Theta^1 e_S,$$

thus

$$y_j^1 = \mathbb{I}(j \in S) + \sum_{k=1}^J \Theta_{jk}^1 \mathbb{I}(k \in S) =: v_j^1(\varnothing)\mathbb{I}(j \in S) + \sum_{k=1:k \neq j}^J v_j^1(\{k\})\mathbb{I}(k \in S),$$

which contains up to first-order interaction.

Now consider $l = 2, \dots, L$. Assume $y^{l-1}$ contains up to $p$-th order interaction such that

$$y_j^{l-1} := v_j^{l-1}(\varnothing)\mathbb{I}(j \in S) + \sum_{\{j_1\} \subseteq S \setminus \{j\}} v_j^{l-1}(\{j_1\})\mathbb{I}(j_1 \in S) + \dots$$

$$+ \sum_{\{j_1,\dots,j_p\} \subseteq S \setminus \{j\}} v_j^{l-1}(\{j_1,\dots,j_p\}) \prod_{i=1}^p \mathbb{I}(j_i \in S).$$

Then the term

$$y_j^l = y_j^{l-1} + \sum_{k=1}^J \Theta_{jk}^l (y_j^{l-1})^2.$$

can contain a high-order term for $j_1' \neq \dots \neq j_p' \neq j_1' \neq \dots \neq j_p'$,

$$\sum_{k=1}^J \Theta_{jk}^l v_j(\{j_1,\dots,j_p\}) v_j(\{j_1',\dots,j_p'\}) \prod_{i=1}^p \big(\mathbb{I}(j_i \in S)\mathbb{I}(j_i' \in S)\big),$$

which is a $2p$-th order interaction. By induction, with $L$ recursion, the model contains up to $2^{L-1}$-th order interactions. Therefore, to represent the full order interaction for $J$ products with up to $J-1$-th order interaction, we need at most $\lceil 1 + \log_2(J-1) \rceil$ recursions.

### A.3 Derivation in Section 4.3

In this section, we derive the identifiability property for the parameter $\alpha$ as defined in (11). Recall $\mathcal{S}$ denotes the full set of products. Suppose for a fixed product pair $\{i, j\} \subseteq \mathcal{S}$, the dataset contains observed market shares for all choice sets in the collection

$$E = \{\{i, j\} \cup B : B \subseteq \mathcal{S} \setminus \{i, j\}\}.$$

We claim that the system of equations defined by (1) is square and full-rank such that the relative halo effect parameters $\{\alpha_{jk}(T)\}_{B \subseteq \mathcal{S} \setminus \{i,j\}}$ are uniquely identifiable from the observed data.

To prove this claim, we need to prove that the number of new variables, denoted as $N_\alpha$, is the same as the number of equations associated with them (from the choice probability of each choice set), denoted as $N_{ep}$, and

$$N_{ep} = \sum_{q=2}^{n} \binom{n}{q} \times (q-1). \tag{13}$$

Note that the summation $\sum_{q=2}^{n} \binom{n}{n-q} = \sum_{q=2}^{n} \binom{n}{q}$ is the number of subset of all items whose size is smaller than $n-2$. These sets can be the source set $T$ of halo effect difference $\alpha$, since there is at least one pair of items out of these sets. Suppose the number of items in the subsets is $q$. We only need $q-1$ halo effect differences to calculate all pair-wise differences within $q$ items. Therefore,

$$N_\alpha = N_{ep} = \sum_{q=2}^{n} \binom{n}{q} \times (q-1). \tag{14}$$

Furthermore, prior work such as Batsell and Polking [1985a] and Park and Hahn [1998b] has shown that the corresponding coefficient matrix is full-rank, as the system represents a saturated model. This guarantees the linear independence of equations, and hence, the parameters $\alpha$ are identifiable.

## B  Implementation Details

### B.1  Featureless Models

**Model Width**  In our formulation (Equation 9 and Equation 10), we initially constrain the model width to match the size of the item universe $J$, setting the shape of $\Theta^l$ to $\mathbb{R}^{J \times J}$. This aligns our model with prior work, such as the low-rank Halo MNL model [Ko and Li, 2024] and contextual MNL [Yousefi Maragheh et al., 2020a].

While a width of $J$ suffices to capture zeroth- and first-order Halo effects, it is inefficient for higher-order interactions. As the number of possible interactions grows exponentially with order, deeper layers must activate and compose increasingly complex patterns from lower-order terms. However, under a fixed width, scaling model capacity necessitates increasing depth, which complicates optimization and training stability.

To alleviate this limitation, we expand the width of the first layer to $\Theta^1 \in \mathbb{R}^{J' \times J}$, and all subsequent layers to $\Theta^l \in \mathbb{R}^{J' \times J'}$ for $l \geq 2$. After the final layer, we project $y^L$ back to dimension $J$ using a linear transformation $W^L \in \mathbb{R}^{J \times J'}$. By choosing $J' > J$, we effectively increase model capacity. We refer to $J'$ as the model width. In essence, this over-parametrizes lower-order representations to better facilitate the modeling of higher-order effects. We refer the readers to more details about this in Appendix A.2.2.

### B.2  Feature-based Models

**Non-Linear Embedding** $\chi$  In our implementation, the non-linear embedding function $\chi$ is defined as a three-layer multilayer perceptron (MLP) with ReLU activations after each hidden layer and a final layer normalization. The network maps the input $x \in \mathbb{R}^{d_x}$ to an embedding space of dimension $d$:

$$\chi(x) = \text{LayerNorm}\left(W_3 \cdot \text{ReLU}\left(W_2 \cdot \text{ReLU}\left(W_1 x + b_1\right) + b_2\right) + b_3\right), \quad x \in \mathbb{R}^{d_x} \tag{15}$$

where $W_1 \in \mathbb{R}^{d \times d_x}$, $W_2, W_3 \in \mathbb{R}^{d \times d}$, and $b_1, b_2, b_3 \in \mathbb{R}^d$. This embedding function is shared by all items in the choice set.

**Head-specific Nonlinear Transformations** $\phi_h^l$  We implement the head-specific non-linear transformation $\phi_h^l$ as a two-layer MLP. The first layer is specific to each head $h$, while the second (output) layer is shared across all heads. A LayerNorm is applied after the second layer to stabilize training.

$$\phi_h^l(z_j^0) = \text{LayerNorm}\left(W_{\text{shared}}^l \cdot \text{ReLU}(W_h^l z_j^0 + b_h^l) + b_{\text{shared}}^l\right), \quad z_j^0 \in \mathbb{R}^d \tag{16}$$

where $z_j^0 = \chi(x_j) \in \mathbb{R}^d$, $W_h^l \in \mathbb{R}^{d \times d}$, $W_{\text{shared}}^l \in \mathbb{R}^{d \times d}$, and $b_h^l, b_{\text{shared}}^l \in \mathbb{R}^d$.

**Dummy Items**    To enable parallel computation and efficient model training, all choice sets are padded to a uniform size. We pad dummy items using zero vectors in the input space, i.e., $\mathbf{0}^{d_x}$, and explicitly enforce their latent representations to remain inactive throughout the network. Specifically, after each residual layer, we reset $Z_j^l = \mathbf{0}^d$ for any dummy item $j$, ensuring that dummy items do not accumulate or propagate any signal across layers. Since $\sigma$ is defined as an element-wise polynomial activation function, it satisfies $\sigma(\mathbf{0}) = \mathbf{0}$, ensuring that the aggregation process Equation 6 remains unaffected by dummy items. In the final softmax mapping, we set all dummy items' utility value to be $-inf$. This mechanism guarantees that dummy items do not affect contextual aggregation or final predictions, thereby preserving the integrity of the learned representations.

## C    Empirical Study Details

### C.1    Featureless Experiments

#### C.1.1    Synthetic Data Experiments

**Data Generation**    In this experiment, we directly sample the choice probability vector uniformly from the 15-dimensional probability simplex $\mathbb{R}^{15}$, rather than sampling each individual halo effect from a distribution—such as the standard normal—and then computing the corresponding choice probabilities. While the latter approach may seem more intuitive or realistic, it would require evaluating the choice probabilities exponentially many times due to the complex structure of the choice sets, making it computationally inefficient. Importantly, any choice data can be represented by an appropriate system of halo effects [Batsell and Polking, 1985b], implying that it is not necessary to explicitly specify the underlying halo effects. This justifies the feasibility and practicality of our direct sampling approach.

**Experiments Setup**    We conduct two sets of experiments under parameter budgets of 200k and 500k, respectively. For each budget, we vary the model depth from 3 to 7. The batch size and learning rate are fixed to be 1024 and $1 \times 10^{-4}$. All models are trained for 500 epochs. The detailed configurations and corresponding training results are summarized below (see Table 4). All experiments, including hypothetical data experiments, are conducted on a single Google Colab T4 GPU with Adam optimizer.

Table 4: Model configurations with depth, width, parameter count, and in-sample training RMSE.

| Group | Depth | Width | Param Count | Training RMSE |
|---|---|---|---|---|
| 200k | 3 | 306 | 200.736k | 0.0434 |
| | 4 | 251 | 200.298k | 0.0243 |
| | 5 | 218 | 200.124k | 0.0202 |
| | 6 | 195 | 199.290k | 0.0188 |
| | 7 | 179 | 200.838k | 0.0183 |
| 500k | 3 | 489 | 499.758k | 0.0419 |
| | 4 | 401 | 500.448k | 0.0156 |
| | 5 | 348 | 500.424k | 0.0140 |
| | 6 | 312 | 501.384k | 0.0131 |
| | 7 | 285 | 501.030k | 0.0130 |

In these experiments, we use the mean squared error (MSE) between the predicted choice probabilities and the ground-truth one-hot choice vectors as the training objective. This allows us to use training RMSE as a proxy for evaluating the model's approximation error. Note that the training RMSE reported in Table 4 and Figure 2 is not computed directly from the loss. Instead, for each choice set, we first compute the empirical choice frequency vector and then measure the RMSE between this vector and the model's predicted choice probabilities. A perfect fit would result in an RMSE of zero. This evaluation method highlights the expressiveness of the model, as it directly reflects how well the predicted distributions capture the underlying variability in observed choices, beyond merely fitting to individual samples.

### C.1.2 Real Dataset Experiments

**Hotel** From our experimental results, the contextual effects are more pronounced in the second hotel, suggesting that user choices in this setting are more strongly influenced by the composition of the choice set. As a result, we select the second hotel as a case study to compare our method against baseline models, in order to better evaluate their ability to capture such contextual dependencies.

**SFOwork and SFOshop** The **SFOwork** dataset includes six transportation modes: driving alone, shared ride (2), shared ride (3+), bike, transit, and walk. The **SFOshop** dataset comprises eight transportation modes: SharedRide (2+) and DriveAlone, SharedRide (2/3+), DriveAlone, SharedRide (3+), walk, transit, SharedRide (2), and bike. These datasets originate from Koppelman and Bhat [2006], and we use the preprocessed versions provided by Seshadri et al. [2019].

Table 5: DeepHalo hyper-parameters and training settings for experiments on Hotel, SFOshop, and SFOwork datasets.

| Hyper-parameter | Hotel | SFOshop | SFOwork |
|---|---|---|---|
| Model Width $J'$ | 3 | 20 | 20 |
| Layers $L$ | 4 | 5 | 5 |
| Polynomial Activation $\sigma$ | Quadratic | Quadratic | Quadratic |
| Batch size | Full Batch | 256 | 256 |
| Learning Rate | $1 \times 10^{-3}$ | $1 \times 10^{-4}$ | $1 \times 10^{-4}$ |

**Experiments Setup** For the HOTEL dataset, due to its small size and the absence of a dedicated validation split, we train the model using a full-batch setting for 300 epochs without early stopping. To mitigate overfitting, we adopt a compact model configuration with reduced width and fewer parameters. In contrast, for experiments on the SFO datasets (SFOSHOP and SFOWORK), we tune the number of layers $L \in \{4, 5\}$ and the intermediate width $J' \in \{10, 20\}$ based on validation performance. Early stopping is applied with a patience of 10 epochs to prevent overfitting. All experiments are conducted on a single Google Colab T4 GPU with Adam optimizer.

**Reproducibility Remark** We observe that across all three datasets, the experimental results are highly stable: the standard deviations of both NLL and accuracy over 5 runs are consistently close to zero. Moreover, top-1 accuracy, while intuitive, shows limited sensitivity in reflecting model performance differences due to the relative simplicity of the tasks. Therefore, for brevity and clarity, we omit the full multi-run result tables from the appendix. We have, however, ensured that all reported findings are representative and reproducible.

### C.2 Feature-based Experiments

#### C.2.1 Datasets Details

**Expedia Dataset** For data preprocessing, we generally follow the procedures outlined in Aouad et al. [2021]. Specifically, we one-hot encode the categorical features `site_id`, `visitor_location_country_id`, `prop_country_id`, and `srch_destination_id`, grouping all categories with fewer than 1,000 occurrences into a special category labeled -1. Continuous features such as `price_usd` and `srch_booking_window` are filtered to remove unrealistic values: we retain only searches with hotel prices between $10 and $1,000 and booking windows shorter than one year. A logarithmic transformation is applied to both features to reduce skewness. Missing values across the dataset are imputed using the placeholder value -1.

We further filter out all records where no hotel was chosen, as such cases are not informative for modeling discrete choice and constitute a large portion of the raw data. To handle dummy items introduced for padding, we assign all input features of these items to zero vectors. After preprocessing, we obtain a dataset consisting of 275,609 transaction observations, each described by 35 item-specific features and 56 shared features.

**LPMC Dataset** The LPMC dataset is preprocessed by constructing both item-specific and item-shared features for the four available transportation modes: walk, cycle, public transit, and drive. Each alternative is represented by a 4-dimensional item-specific feature vector that includes duration

(e.g., walking time, cycling time, or transit access/rail/bus/interchange time for public transit), cost (such as fuel prices, transit fare, or congestion charges), number of interchanges (for transit), and a congestion level indicator (for driving).

In addition, we incorporate a set of shared features that are common across all items within a choice set. These include numerical variables such as straight-line distance, user age, gender (binary), driver's license status (binary), and the number of cars owned. Categorical variables like day of the week and trip purpose are encoded using one-hot representations. After preprocessing, the final dataset includes 8 item-specific features and 17 shared features for each sample.

### C.2.2 Experiments Details

**Experiments Setup**  We tune the hyperparameters of DeepHalo based on validation performance. Specifically, we search the number of layers $L \in \{4, 5\}$, embedding dimensions $d \in \{32, 64\}$, and hidden dimensions $H \in \{8, 16\}$. For all configurations, we adopt early stopping with a patience of 10 epochs to prevent overfitting. On the LPMC dataset, we further employ a learning rate scheduling strategy: training is first conducted with a learning rate of $10^{-3}$, and then fine-tuned using a smaller rate of $10^{-4}$. This two-phase training helps improve convergence stability and final model performance. More details are shown in Table.6. All experiments are conducted on a single Google Colab T4 GPU with Adam optimizer.

Table 6: DeepHalo hyper-parameters and training settings on Expedia and LPMC Experiments.

| Hyper-parameter | Expedia | LPMC |
|---|---|---|
| Embedding dimension $d$ | 32 | 32 |
| Hidden size $H$ | 8 | 8 |
| Layers $L$ | 5 | 4 |
| Polynomial Activation $\sigma$ | Hadamard product | Hadamard product |
| Batch size | 256 | 256 |
| Learning Rate | $1 \times 10^{-3}$ | $1 \times 10^{-3}/1 \times 10^{-4}$ |

**Baseline Information**  We summarize below the key configurations and model architectures for each baseline used in our experiments.

*TCNet* adopts a single-layer Transformer encoder-decoder architecture with multi-head self-attention and feedforward layers. Both the source and target inputs are linear projections of item features. For the LPMC dataset, we use an embedding dimension of $d = 36$ and $K = 6$ attention heads; for the Expedia dataset, we set $d = 64$ and $K = 8$. The decoder output for each item is passed through a final linear layer to obtain utility scores.

*RUMnet* models sequential utility construction using a GRU that processes items in their presented order within each choice set. At each timestep, the GRU updates a latent preference state, from which the utility for the current item is computed using a linear transformation. Additionally, an item-specific bias term, learned from raw features, is added to the final score. The hidden size of the GRU is set to 96 for LPMC and 128 for Expedia to accommodate the respective feature complexities.

*TasteNet* implements a latent factor model with learned item and user representations. Both the item encoder and the user encoder are two-layer multilayer perceptrons (MLPs) with a hidden size of 128 and output dimension 128. The user embedding is obtained by averaging valid item feature vectors within a choice set. Final utilities are computed as dot products between item and user embeddings, with an item-specific bias added.

*FATEnet* uses a DeepSet architecture where the embedding dimension matches the input size $d_x$. Both the embedding network and the pairwise utility network are implemented as five-layer MLPs with 64 hidden units per layer and ReLU activations. The pairwise utility network outputs a scalar utility score for each item-context pair.

*MLP* is a simple three-layer MLP with ReLU activations and a default hidden width of $d_{\text{embed}} = 128$. It maps each item feature vector to a scalar utility, followed by a masked softmax normalization over available alternatives. The model does not include any residual connections or context aggregation mechanisms.

*ResLogit* encodes each item through a three-layer MLP with embedding dimension $d = 32$, using ReLU activations and dropout. The encoded features are passed through a layer normalization layer and then linearly projected via a learned coefficient vector $\beta \in \mathbb{R}^d$. The resulting utility scores are refined through a 10-layer residual network consisting of nonlinear residual blocks based on softplus activations, followed by masked softmax.

*DLCL* is a context-aware linear model in which item utilities are computed via a learned matrix $B$ modulated by context-dependent adjustments from a second matrix $A$, based on the average feature vector of the entire choice set. The outputs from different feature dimensions are combined using a set of learnable mixture weights to produce the final utility distribution.

**Detailed Results** We run each experiment with different random seeds and report the mean and standard deviation of the evaluation metrics from 5 repetitions. This improves statistical robustness and accounts for variance due to initialization and stochastic training dynamics. We evaluate model performance using negative log-likelihood (NLL) and top-1 accuracy, where accuracy is defined as the proportion of cases where the model assigns the highest predicted probability to the actually chosen item. Unless otherwise specified, the main results reported in the paper correspond to a single representative run. The summary of detailed results is shown in Table 7 and Table 8.

Table 7: Expedia Detailed Results

| Model | Train | | Validation | | Test | |
|---|---|---|---|---|---|---|
| | Loss | Acc | Loss | Acc | Loss | Acc |
| DeepHalo | 2.5086 ± 0.0051 | 0.2501 ± 0.0011 | **2.5216 ± 0.0025** | **0.2474 ± 0.0007** | **2.5288 ± 0.0026** | **0.2442 ± 0.0009** |
| TCnet | **2.4965 ± 0.0125** | **0.2519 ± 0.0034** | 2.5260 ± 0.0051 | 0.2476 ± 0.0006 | 2.5343 ± 0.0041 | 0.2423 ± 0.0005 |
| MLP | 2.5693 ± 0.0030 | 0.2362 ± 0.0008 | 2.5695 ± 0.0011 | 0.2371 ± 0.0009 | 2.5805 ± 0.0015 | 0.2304 ± 0.0011 |
| FATEnet | 2.5460 ± 0.0067 | 0.2407 ± 0.0015 | 2.5440 ± 0.0046 | 0.2421 ± 0.0008 | 2.5548 ± 0.0057 | 0.2365 ± 0.0008 |
| ResLogit | 2.5540 ± 0.0044 | 0.2401 ± 0.0015 | 2.5577 ± 0.0015 | 0.2396 ± 0.0011 | 2.5692 ± 0.0028 | 0.2342 ± 0.0008 |
| RUMnet | 2.5565 ± 0.0041 | 0.2397 ± 0.0016 | 2.5685 ± 0.0011 | 0.2358 ± 0.0009 | 2.5782 ± 0.0026 | 0.2308 ± 0.0021 |
| TasteNet | 2.5623 ± 0.0081 | 0.2383 ± 0.0016 | 2.5675 ± 0.0036 | 0.2374 ± 0.0007 | 2.5766 ± 0.0040 | 0.2326 ± 0.0008 |
| DLCL | 2.5624 ± 0.0009 | 0.2357 ± 0.0002 | 2.5546 ± 0.0010 | 0.2391 ± 0.0005 | 2.5621 ± 0.0010 | 0.2307 ± 0.0002 |
| MNL | 2.6272 ± 0.0001 | 0.2260 ± 0.0002 | 2.6160 ± 0.0001 | 0.2275 ± 0.0002 | 2.6245 ± 0.0001 | 0.2210 ± 0.0003 |

Table 8: LPMC Detailed Results

| Model | Train | | Validation | | Test | |
|---|---|---|---|---|---|---|
| | Loss | Acc | Loss | Acc | Loss | Acc |
| DeepHalo | **0.6427 ± 0.0122** | **0.7527 ± 0.0037** | **0.6412 ± 0.0053** | **0.7576 ± 0.0028** | **0.6407 ± 0.0045** | **0.7552 ± 0.0022** |
| TCnet | 0.6744 ± 0.0098 | 0.7422 ± 0.0036 | 0.6581 ± 0.0096 | 0.7500 ± 0.0038 | 0.6577 ± 0.0076 | 0.7468 ± 0.0023 |
| MLP | 0.7036 ± 0.0034 | 0.7333 ± 0.0019 | 0.6855 ± 0.0029 | 0.7402 ± 0.0019 | 0.6870 ± 0.0049 | 0.7367 ± 0.0030 |
| FATEnet | 0.6765 ± 0.0042 | 0.7413 ± 0.0015 | 0.6583 ± 0.0046 | 0.7508 ± 0.0018 | 0.6619 ± 0.0043 | 0.7457 ± 0.0029 |
| ResLogit | 0.7080 ± 0.0047 | 0.7293 ± 0.0025 | 0.6926 ± 0.0057 | 0.7348 ± 0.0022 | 0.6915 ± 0.0052 | 0.7370 ± 0.0017 |
| RUMnet | 0.6923 ± 0.0074 | 0.7378 ± 0.0017 | 0.6754 ± 0.0079 | 0.7460 ± 0.0025 | 0.6779 ± 0.0067 | 0.7411 ± 0.0020 |
| TasteNet | 0.7140 ± 0.0056 | 0.7292 ± 0.0031 | 0.6976 ± 0.0056 | 0.7356 ± 0.0032 | 0.6963 ± 0.0068 | 0.7348 ± 0.0035 |
| DLCL | 0.7148 ± 0.0027 | 0.7151 ± 0.0013 | 0.6978 ± 0.0016 | 0.7266 ± 0.0018 | 0.6988 ± 0.0026 | 0.7218 ± 0.0011 |
| MNL | 0.8813 ± 0.0000 | 0.6312 ± 0.0002 | 0.8621 ± 0.0000 | 0.6426 ± 0.0003 | 0.8637 ± 0.0001 | 0.6441 ± 0.0002 |

## C.3 Additional Experiments

### C.3.1 Data Efficiency on the SFOshop Dataset

To evaluate data efficiency, we subsample the SFOshop dataset—known for strong context effects—at ratios from 10% to 100%, training all neural models under a strict parameter budget of approximately 1K. Across all settings, DeepHalo consistently achieves the lowest test NLL, showing strong generalization even with limited data. Its test NLL steadily improves with more data, while other models (TCNet, MLP, FATE) either plateau early or fluctuate.

### C.3.2 Training Time Comparison

Table 10 reports total training time (until early stop) in a representative LPMC setting. DeepHalo requires longer training due to explicit order-decomposition, but remains within the same computational scale as other deep baselines.

Table 9: Data efficiency comparison on SFOshop (test NLL). Lower is better.

| Ratio (%) | CMNL | MNL | FATE | MLP | MixedMNL | TCNet | DeepHalo |
|---|---|---|---|---|---|---|---|
| 10 | 1.574 | 1.898 | 1.577 | 1.571 | 1.674 | 1.589 | 1.566 |
| 20 | 1.567 | 1.869 | 1.576 | 1.563 | 1.617 | 1.573 | 1.558 |
| 30 | 1.564 | 1.844 | 1.574 | 1.557 | 1.590 | 1.569 | 1.551 |
| 40 | 1.562 | 1.820 | 1.574 | 1.554 | 1.576 | 1.568 | 1.548 |
| 50 | 1.561 | 1.779 | 1.576 | 1.550 | 1.566 | 1.570 | 1.544 |
| 60 | 1.562 | 1.760 | 1.576 | 1.545 | 1.563 | 1.551 | 1.537 |
| 70 | 1.555 | 1.742 | 1.576 | 1.542 | 1.560 | 1.539 | 1.532 |
| 80 | 1.544 | 1.726 | 1.576 | 1.537 | 1.558 | 1.560 | 1.528 |
| 90 | 1.550 | 1.699 | 1.576 | 1.539 | 1.555 | 1.549 | 1.526 |
| 100 | 1.550 | 1.726 | 1.560 | 1.537 | 1.558 | 1.538 | 1.528 |

Table 10: Training time comparison under the LPMC setting.

| Model | Total Training Time (s) |
|---|---|
| MNL | 96.5 |
| MLP | 32.3 |
| TasteNet | 45.5 |
| RUMNet | 102.2 |
| DLCL | 298.1 |
| ResLogit | 143.3 |
| FATE | 61.3 |
| TCNet | 141.6 |
| DeepHalo | 202.0 |

### C.3.3 Featureless Dataset Baselines

We additionally evaluate FATE, TCNet, and MixedMNL in featureless settings to ensure a fair comparison. All neural models are controlled to approximately 1K parameters to focus on architectural differences.

Table 11: Results on featureless datasets (Train/Test NLL). Lower is better.

| Model | Hotel | SFOshop | SFOwork |
|---|---|---|---|
| MNL | 0.7743 / 0.7743 | 1.7281 / 1.7262 | 0.9423 / 0.9482 |
| MLP | 0.7569 / 0.7523 | 1.5556 / 1.5523 | 0.8074 / 0.8120 |
| CMNL | 0.7566 / 0.7561 | 1.5676 / 1.5686 | 0.8116 / 0.8164 |
| MixedMNL | 0.7635 / 0.7613 | 1.5599 / 1.5577 | 0.8092 / 0.8153 |
| FATE | 0.7575 / 0.7568 | 1.5726 / 1.5765 | 0.8133 / 0.8167 |
| TCNet | 0.7467 / 0.7670 | 1.5694 / 1.5742 | 0.8114 / 0.8145 |
| DeepHalo (ours) | 0.7479 / 0.7483 | 1.5385 / 1.5263 | 0.8040 / 0.8066 |

