# OpenReview forum: "DeepHalo: A Neural Choice Model with Controllable Context Effects"
_NeurIPS.cc/2025/Conference — NeurIPS 2025 spotlight_

### Official Review · Reviewer_D2h9 · 2025-06-13

**Clarity:** 3
**Significance:** 3
**Originality:** 3
**Rating:** 4
**Confidence:** 4

**Summary:**

This paper introduces DeepHalo, a neural modeling framework designed to capture context-dependent choice behavior. Traditional choice models can fail to capture dependent utilities while recent context-dependent choice models are either restricted by interaction structures or lacking interpretability.  The paper first shows a utility decomposition with features and then introduces DeepHalo to capture decomposed higher-order effects. This novel architecture can bridge the previous gap by providing an expressive model that can both explicitly control the complexity of interactions (i.e., the order of interactions) and offer a principled interpretation of these context effects. The paper provides empirical validation on synthetic and real-world datasets (both featureless and feature-based), showing that DeepHalo achieves competitive or superior predictive performance compared to existing baselines while offering enhanced interpretability.

**Questions:**

1. Could you provide the number of parameters for each benchmark model used in your experiments?
2. The formulation of higher-order interactions in Equations (4) and (5) appears conceptually similar to the transformer architecture, with the key difference being that the attention mechanism is replaced by a linear transformation followed by an activation function (just like assigning equal attention across elements). Could you elaborate on the differences between your approach and transformers, both in terms of formulation and implications for model expressiveness or computational efficiency?

**Ethical Concerns:**

["NO or VERY MINOR ethics concerns only"]

**Final Justification:**

Please see Strengths And Weaknesses.

**Limitations:**

See weaknesses.

**Quality:**

3

**Strengths And Weaknesses:**

Strengths: The paper is generally very well-written and organized. While building upon established concepts like utility theory, permutation equivariance, and residual connections, the specific architecture of DeepHalo that aims to explicitly model and disentangle interactions layer-by-layer is novel and insightful. Its interpretability is very useful when we need to learn the users preferences in practice.

Weaknesses:
1) In experiments, I think it is essential to provide the number of parameters of each benchmark model to make sure the better performance of DeepHalo is from the design instead of the size.
2) While many benchmark models are compared in datasets with features (Table 2), while in featureless datasets (Table 1), only a few models are tested. However, I think the models like FateNet, TCNet can also be applied to featureless cases.

Minor comments:
1) It would be helpful if a figure can be provided to illustrate the architecture.
2) Typo: Line 79 "Deep Sets [Zaheer et al., 2018], Deep Sets [Zaheer et al., 2018]"
3) While the std are provided in Appendix (e.g., Table 6, 7), it would be helpful if you also provide them in the main part (e.g., Table 1,2,).

---

> ### Author Rebuttal · Authors · 2025-07-31
>
> We thank Reviewer D2h9 for their positive recognition of our work. In the revision, we will include a model architecture diagram and correct the noted typos. Below, we provide point-by-point responses to the reviewer’s comments.
>
>
> ### Responses to Weakness
>
> > Q: In experiments, I think it is essential to provide the number of parameters of each benchmark model to make sure the better performance of DeepHalo is from the design instead of the size.
>
> A: We thank the reviewer for raising the question regarding the model scale. To address this, we computed the number of trainable parameters under the **Expedia experiment setting**, where the input item feature dimension is set to 10 across all models. The results are summarized below:
>
> | Model      | Number of Parameters |
> |------------|----------------------|
> | TCNet      | 118,209              |
> | **DeepHalo**   | **53,833**               |
> | RUMNet     | 53,900               |
> | TasteNet   | 35,851               |
> | DLCL       | 210                  |
> | FATE       | 44,363               |
> | MLP        | 42,601               |
> | ResLogit   | 2,720                |
> | MNL        | 10                   |
>
> > Q: While many benchmark models are compared in datasets with features (Table 2), while in featureless datasets (Table 1), only a few models are tested. However, I think the models like FateNet, TCNet can also be applied to featureless cases.
>
> A: We thank the reviewer for their suggestion. In the featureless setting, we have added results for three representative baselines: FATE, TCNet, and Mixed MNL. To ensure fairness and focus on architectural differences, we controlled the parameter size of all neural network–based models to be **approximately 1K parameters**. The results are shown below:
>
> | Model         | Hotel (Train/Test) | SFOshop (Train/Test) | SFOwork (Train/Test) |
> |---------------|--------------------|------------------------|-----------------------|
> | MNL           | 0.7743 / 0.7743    | 1.7281 / 1.7262        | 0.9423 / 0.9482       |
> | MLP           | 0.7569 / 0.7523    | 1.5556 / 1.5523        | 0.8074 / 0.8120       |
> | CMNL          | 0.7566 / 0.7561    | 1.5676 / 1.5686        | 0.8116 / 0.8164       |
> | Mixed MNL     | 0.7635 / 0.7613    | 1.5599 / 1.5577        | 0.8092 / 0.8153       |
> | FateNet       | 0.7575 / 0.7568    | 1.5726 / 1.5765        | 0.8133 / 0.8167       |
> | TCNet         | **0.7467** / 0.7670| 1.5694 / 1.5742        | 0.8114 / 0.8145       |
> | **DeepHalo** (Ours) | 0.7479 / **0.7483** | **1.5385** / **1.5263** | **0.8040** / **0.8066** |
>
> ### Responses to Questions
>
> > Q: The formulation of higher-order interactions in Equations (4) and (5) appears conceptually similar to the transformer architecture, with the key difference being that the attention mechanism is replaced by a linear transformation followed by an activation function (just like assigning equal attention across elements). Could you elaborate on the differences between your approach and transformers, both in terms of formulation and implications for model expressiveness or computational efficiency?
>
> A: This is a highly insightful observation. Indeed, the connection to Transformer-style architectures is deeply embedded in the design of our approach, and we appreciate the opportunity to elaborate on this.
>
> In summary, there are two differences between DeepHalo and Transformers, both structurally and in modeling implications:
>
> - **Effect order control**:
>   Transformers (e.g., in TCNet) implicitly allow interactions across all items, inevitably resulting in **full-order interactions** (order $|S| - 1$). In contrast, DeepHalo provides explicit layer-wise control, where each layer adds exactly one level of interaction (or $\times 2$ with quadratic activation). This control is essential for interpretability, regularization, and tractability.
>
> - **Structural simplification**:
>   DeepHalo eliminates the need for query/key/value projections and softmax attention, replacing them with fixed linear transformations and polynomial activations. This results in lower computational cost—linear in $|S|$, rather than quadratic—as well as a simpler and more transparent architecture.
>
> Below, we briefly explain the motivation behind our model design to provide a more intuitive understanding of our model architecture.
>
> We begin with the **featureless setting**, where modeling higher-order context effects reduces to polynomial regression over indicator functions that encode item presence. DeepHalo introduces a structured and recursive mechanism to capture **subset-level interactions** in this setting, conceptually analogous to the accumulation of interactions based on binary inclusion indicators.
>
> In the **feature-rich setting**, the core challenge is to encode interactions among a subset of items using their feature representations, while strictly excluding information from outside the subset. This naturally recalls the concept of attention scores, typically computed from item feature pairs. However, when these scores are normalized via a softmax function, **the resulting attention weights entangle all item features**, thus violating the independence required for controlled halo modeling.
>
> For instance, when evaluating the influence of subset $\{A, B\}$ on the utility of item $C$ within the offer set $\{A, B, C, D\}$, the effect should depend solely on the features of $A$, $B$, and $C$, and remain invariant to unrelated items such as $D$. The softmax operation introduces information from all items into the denominator of each attention weight, which naturally violates this principle.
>
> To address this, we omit softmax normalization, and further observe that the dot-product attention mechanism is not essential for modeling decomposable utility functions. This insight motivates a simplification of the architecture, leading to the formulations presented in Equations (4) and (5).
>
> In short, **while DeepHalo draws conceptual inspiration from Transformer-style aggregation, it diverges in architectural design to achieve interpretable, order-controllable, and computationally efficient modeling of contextual choice effects**. We thank the reviewer again for raising this connection and will add a discussion in the final version to highlight these points explicitly.

---

> > ### Comment · Reviewer_D2h9 · 2025-08-05
> >
> > Thank you for the clarifications and additional experiments. I maintain my positive score for this work.

---

### Official Review · Reviewer_6ydQ · 2025-06-17

**Clarity:** 3
**Significance:** 2
**Originality:** 3
**Rating:** 5
**Confidence:** 5

**Summary:**

This paper introduces an ML based context-dependent choice model. The issue is that the classical IIA assumption of choice models is often not met/questionable, and one perspective on addressing it is through modeling the “halo” effect. In this approach the authors consider using information from all alternatives when evaluating each single one, and considering their interactions. The paper shows the derivations of these interactions in a permutation equivariant deep neural network framework. Results show that the model generally outperforms the baselines.

**Questions:**

I enjoyed reading the paper, and this seems lile an interesting research direction. However, it’s very unclear to me why this fits in the NeurIPS context. The methodological ML contribution lies practically exclusively in the modeling of interactions and equivariance, and in both cases, the contributions are not very novel. If the contribution relates to DCM behaviour modeling for the ML community, then some aspects should be considered (e.g. why not consider nested logit, mixed logit, LCCM? why classical MNL as baseline, which is the same for DCM literature as fully connect MLP is for deep learning (the most vanilla of vanilla…).
Don’t take me wrong, I do think this is an interesting paper, but I’d see it as more appropriate for Journal of Choice Modeling for example…

**Ethical Concerns:**

["NO or VERY MINOR ethics concerns only"]

**Final Justification:**

The authors addressed all my core concerns and improved/will improve the paper substantially.

**Limitations:**

Some clarifications:

- The intro is good but some things are a bit unclear. For example, “featureless settings”?? What is that (I know what it is, but I did have my doubts in the beginning, and it would be nice to spare the reader of such trivial doubts)?

- “For each j ∈S, define a utility function uj(T), which represents the utility of alternative j when the context is the subset T ⊆S\{j}.” This is quite confusing. You are introducing the classical MNL, but also introduce the concept of context at the same time. Fine, but then, you refer to uj(T) (utility of j within the context?) in the text, while the MNL presented has uj(S). What’s the relationship here?

- “Here, XT∪{j}∈Rdx ×J denotes the matrix formed by replacing the feature vectors not in the subset
T ∪{j}with zero.” —> rephrase. At first sight, it almost implies that this matrix is only zeros (it’s not mentioned what to do with the features that ARE in the subset). I know this is obvious, but the cognitive load of a reader would thank the authors…

- This sentence got me a bit confused: “Here, H denotes the number of interaction heads, analogous to the number of channels or attention heads, and thereby controls the diversity of the interaction patterns. “ It seems as though W^1 is projecting the z embeddings vectors to a (lower?) space, and that W^1 is shared by all alternatives (there is ONE W^1). Then, why not train the z with dimensionality h directly? What’s the benefit of W^1?

-Section “4.1 Residual Connection for Large Choice Sets” is quite confusing: in what sense does polynomial aggregation improve computational efficiency? In fact, with larger S, then you will get extremely high degree polynomials, which will be prone to numerical errors. Perhaps the benefit is then that you can use polynomials to reduce number of layers (i.e. trade-off linearity with many layers with polynomials with less layers), but this is not clear in the explanation nor on the notation.

- For section 4.2., if I understand well, e_S is a vector of 1s. If that is true, then u_j(e_S) is a sum of ones irrespeective of the polynomials, and thus totally S. This makes no sense, implying I’m misunderstanding something here… I do understand that, later on, given that we learn the q values (eq 9), this no longer applies (i.e. we get polynomials), and it makes intuitive sense.


- There’s something confusing with the meaning of \union in equation 3. Classical interpretation implies that T \union {j} is just the set T plus the element j. But It sort of implies that j is treated differently to the rest. Ok, you already index v_j (on the left of the equation), thus allowing you to differentiate j on the right hand side. At first time, that was totally fine. But then we get to equation 10, in particular v_j(T) + vj_(T \union {k}), where you make vj_(T \union {k}), which becomes ambiguous. I mean, I understand that you want to estimate the context effect of T and also consider the context role of k in interaction with T, but then what would be expansion (according to 3)?

- About number of parameters in Fig 2, what are we talking about specifically? Dimensionality of embeddings? Of the phi functions?

- \bar{Z}_1 := 1/S…  I believe it should be |S|, since S is a set.


The beverage experiment is not very intuitive. The strongest effect seems to be that, if Pepsi is on the menu, then people strongly prefer 7Up or Sprite rather than Coke (versus having only 3 options? the entire set? Versus having a different cola instead of pepsi? Is the choice set the competing product pair+effect source set in each case? Is this featureless? What are the features?). Why? What’s the intuitive explanation? The examples in the intro/motivation section were very convincing, why not try them instead?

**Paper Formatting Concerns:**

The format seems ok to me.

**Quality:**

4

**Strengths And Weaknesses:**

Strengths:
- Important modeling problem (although somewhat specific/niche)
- Convincing methodological treatment, with only a few notation glitches
- Successful implementation and validation with real data
- Good choice of (ML related) baselines

Weaknesses:
- Poor choice of (classical DCM) baselines
- Lack of training (computational) performance statistics. How fast/slow is it, compared with the other DNNs?
- Some points are unclear (see below)

---

> ### Author Rebuttal · Authors · 2025-07-31
>
> Thank you very much for your thoughtful comments. Please see our response as follows.
>
> **Weakness & Questions:**
> > **Q: The methodological ML contribution ... the contributions are not very novel.**
>
> A: The contribution of DeepHalo lies not merely in modeling interactions or equivariance, but in introducing a **structured and order-controllable framework** . While prior models have explored interaction effects, they face fundamental challenges:
>
> - First-order models such as CMNL [Yousefi Maragheh et al., 2020], low-rank Halo MNL [Ko and Li, 2024], and FETA [Pfannschmidt et al., 2022] are **inherently restricted to pairwise interactions**. For instance, FETA proposes a decomposition that resembles the halo effect, but practically, it is limited to first-order pairwise interactions. The architecture is not scalable to higher-order effects, as doing so would require exponentially large tensor storage and computational resources, rendering it impractical for real-world applications.
>
> - Other higher-order models, such as FATE [Pfannschmidt et al., 2022], ResLogit [Wong and Farooq, 2021], and TCNet [Wang et al., 2023], attempt to capture context effects by **entangling all item features** in the offer set. While their outputs can be post hoc interpreted as halo effects via Eq. (10), the interaction order is uncontrolled, and recovering a meaningful decomposition requires $O(2^{|S|-1})$ complexity due to exponentially many terms. This stems from their architectures: context is not modeled in a structured or decomposable way. For instance, FATE compresses all item features into a set embedding, while TCNet’s softmax-based attention mixes information across items. As a result, none can recover a truly decomposable utility function. We refer the reviewer to Appendix A2.2, A2.3, and Question 2 of our response to reviewer Unm8 for details on how DeepHalo addresses this limitation.
>
> In contrast, DeepHalo is explicitly designed to address this gap by enabling **precise control over the maximum order of halo interactions**. In e-commerce applications like recommendation and bundle pricing, lower-order context effects are often the most relevant—being more stable, interpretable, and practically useful. While expressive models can recover such effects, they do so implicitly within entangled high-order interactions, requiring up to $O(2^{|S|-1})$ terms to isolate. DeepHalo explicitly models limited-order halo effects, allowing users to control the maximum order $K$ with only $O(2^K)$ complexity, enabling a flexible trade-off between efficiency and expressiveness.
>
>
> > **Q: If the contribution relates to DCM behaviour modeling for the ML community, then some aspects should be considered... (Poor choice of (classical DCM) baselines)**
>
> A: Since our model targets context effects, we primarily compare DeepHalo with context-dependent DCMs like Contextual MNL. MNL serves as a sanity check to highlight the role of context. To include context-independent RUM-based models, we add Mixed Logit in the featureless setting, as it can approximate any RUM. For ML baselines, we include TCNet and FATE (suggested by reviewer D2h9). We respectfully refer the reviewer to our detailed response to Reviewer D2h9.
>
> > **Q: Lack of training (computational) performance statistics.**
>
> A: Thanks for pointing this out. Below, we add the total training time (until early stop) in a sample run under the setting of the LPMC experiment in Section 5.2.
> | Model Name | Total Training Time (s) |
> |------------|-------------------------------|
> | MNL        | 96                          |
> | MLP        | 32.3                          |
> | TasteNet   | 45.5                          |
> | RUMNet     | 102.2                         |
> | DLCL       | 298.1                         |
> | ResLogit   | 143.3                         |
> | FATE       | 61.3                          |
> | TCnet      | 141.6                         |
> | DeepHalo   | 202.0                         |
>
> > **Q: However, it’s very unclear to me why this fits in the NeurIPS context.**
>
> A:  We believe this paper is well-suited for the NeurIPS community, aligning with its growing focus on **human-AI alignment** and **feedback modeling**. Understanding and formalizing human choice behavior is essential for building AI systems that interact reliably with people. Existing alignment methods (e.g., RLHF, LfD) often assume rational, unbiased feedback—assumptions that rarely hold in practice. Our work introduces a principled, interpretable framework for modeling **context-dependent human choices**, laying the foundation for more realistic and user-aligned AI.
>
> Additionally, we list below several noteworthy works related to our topic from previous years’ NeurIPS:
>
> [1]Oh S, Thekumparampil KK, Xu J. Collaboratively learning preferences from ordinal data. Advances in Neural Information Processing Systems. 28, 2015
>
> [2]Stephen Ragain and Johan Ugander. Pairwise choice markov chains. Advances in Neural Information Processing Systems, 29, 2016
>
> [3]Arjun Seshadri, Stephen Ragain, and Johan Ugander. Learning rich rankings. Advances in Neural Information Processing Systems, 33:9435–9446, 2020
>
> [4]Yiqun Hu, David Simchi-Levi, and Zhenzhen Yan. Learning Mixed Multinomial Logits with Provable Guarantees. Advances in Neural Information Processing Systems, 9447-9459, 2022
>
> [5]De Peuter S, Zhu S, Guo Y, Howes A, Kaski S. Preference learning of latent decision utilities with a human-like model of preferential choice. Advances in Neural Information Processing Systems, 37:123608-36, 2024
>
>
> **Clarifications:**
> > **Q: Some things are a bit unclear in intro.**
>
> A: We will explain the ‘Featureless setting’ further by elaborating on ‘items without access to their features’.
>
> > **Q: “For each j ∈S, define a utility function uj(T), ...”**
>
> A: We will only introduce the utility function $u_j(S)$ instead, and only introduce the notation $T$, which emphasizes the context information (subset excluding the item j itself).
>
> > **Q: “Here, $XT∪{j}∈R^{d_x ×J}$ denotes the matrix formed by replacing the feature vectors not in the subset T ∪{j}with zero.” —> rephrase.**
>
> A: We will rewrite the definition of $X_{T∪{j}}$ as “formed from the feature matrix $X_S$ by replacing the feature vectors not in the subset T∪{j} with zero.”
>
> > **Q: This sentence got me a bit confused: “Here, H denotes the number of interaction heads...”**
>
> A: Analogous to multi-head attention, in our context, this can be interpreted as H different consumer types with different tastes. The h-th row of W^1 represents the h-th taste vector, which can map the input with various preferences. Thus, W^1 provides H variety of mappings, which are then aggregated by the sum over $Z_h^1$. This is indeed training each of the h-th mapping, and we use matrix W^1 to compact the notation.
>
> > **Q: “Residual Connection for Large Choice Sets” is quite confusing**
>
> A: As the reviewer noted, one key computational advantage of the polynomial formulation is avoiding the need for deep architectures (line 234). We elaborate on this in Section 4.2 (lines 262–264), showing a logarithmic relationship: a linear number of layers can express exponentially many interaction orders. We will clarify this in the revised version. See also Appendix A2.2, A2.3, and the example in our response to reviewer Unm8 (Section 2).
>
> As for numerical stability, training remains stable in the featureless setting even with high-order interactions (Figure 2). In the feature-rich setting, deep quadratic stacks may cause instability, but we find that modeling up to 4th or 5th order suffices in practice (e.g., Expedia), striking a good balance between expressiveness and interpretability.
>
> > **Q: For section 4.2., $e_S$ is a vector of 1s...**
>
> A: Regarding Section 4.2: $e_S$ is a unit vector with the $(e_S)_j = 1$ if j ∈ S and 0 otherwise. The recursive expression in equation 9 can be represented as a polynomial of $e_S$. Let us consider the case with up to second-order interaction, then the second-order term relates to $e_S \otimes e_S$ where the (i,j)-th element refers to whether both item i and item j exist in the choice set. The coefficients in the polynomial corresponding to each element of the second-order term measure the specific interactions in equation (1).
>
> > **Q: confusing with the meaning of $\cup$ in equation 3.**
>
> A: We apologize for the typo in equation 3: $X_R$ should be $X_{R∪{j}}$, which hopefully entangles the confusion. In equation (10), the relative context effect is specified on item j and item k, whose definition is introduced in Park and Hahn [1998].
>
> > **Q: About number of parameters in Fig 2...**
>
> A: The number of nn parameters in Fig 2 is the total number of trainable parameters, which specifies the model complexity and is influenced by the problem size and the selection of hyperparameters like H.
>
> > **Q: $\bar{Z}_1 := 1/S$…**
>
> A: Yes, thanks for catching that. We will correct to |S|.
>
>
> **Beverage Experiment**
>
> Thank you for the thoughtful comment.
>
> **This classical hypothetical dataset has been widely used in the literature on halo effects** (e.g., Park and Hahn [1998], Batsell and Polking [1985]) for its simplicity and clarity, making it a standard example for evaluating interpretability. The experiment is conducted in a featureless setting, demonstrating that DeepHalo can extract meaningful interaction effects purely from set-level preferences.
>
> Subfigure (b) illustrates how different item subsets (horizontal axis) influence the utility gap between item pairs (vertical axis). The rightmost column (∅) shows baseline preferences without contextual influence—for example, strong preferences for Pepsi over Coke and 7-Up over Sprite are evident both here and in the market shares (Subfigure a). The leftmost column (item 1) reveals a strong negative halo effect of Pepsi on Coke, reducing Coke’s competitiveness when co-offered. A similar pattern is observed between 7-Up and Sprite.

---

> > ### Comment · Reviewer_6ydQ · 2025-08-02
> >
> > Thank you so much for the detailed response. I am happy with the general technical feedback, especially if the authors review the paper accordingly for the final version.
> >
> > However, I still see this paper as a bit of an ambivalent contribution:
> > - On the ML side, it is a sophisticated ellaboration of earlier works (especially Seshadri et al, 2019). I was puzzled by the claim that the paper suits NeurIPS due to 'its growing focus on human-AI alignment and feedback modeling,' as this is only briefly mentioned in the introduction and not even included among the keywords. The issue is that, if this were truly the paper's contribution, it should have been reviewed by experts in those areas. For example, the paper does not provide any actual modeling or empirical study in those domains. The actual contribution lies squarely within context-sensitive choice modeling; the alignment framing appears speculative and unsupported by the substance of the work. For example, 'Human-AI alignment' typically refers to value alignment and ethical decision-making, not behavior modeling—if anything, a powerful choice model could be used manipulatively, which complicates the alignment narrative.
> > - On the DCM side, it is somewhat surprising that the authors overlook nested logit (and cross-nested or mixed probit) formulations, which are the “go-to” solutions for IIA challenges, including with ML methods (e.g. (Aboutaleb et al, 2020)). See some (among many) references below.
> >
> > Aboutaleb, Y. M., Ben-Akiva, M., & Jaillet, P. (2020). Learning nested logit models from choice data. arXiv preprint arXiv:2008.08048. https://arxiv.org/abs/2008.08048
> >
> > McFadden, D. (1978). Modeling the choice of residential location. In A. Karlqvist, L. Lundqvist, F. Snickars, & J. W. Weibull (Eds.), Spatial interaction theory and planning models (pp. 75–96). North-Holland.
> >
> > Mokhtarian, P. L. (2016). The role of travel behavior research in reducing the carbon footprint: A skeptical view. Transportation Research Part D: Transport and Environment, 44, 14–25. https://doi.org/10.1016/j.trd.2015.12.004
> >
> > In conclusion, to be very clear: I think this is an excellent work and I’m not totally against raising the score, but to be truly groundbreaking, it should not pass as a niche contribution in NeurIPS—effectively bypassing reviewers from the core discrete choice modeling community. To raise my score, I want to get the clear feeling that this paper is not just “yet another data-driven sophisticated MNL that deals with IIA without using nested logit or other typical parametric models”.

---

> > > ### Author Response · Authors · 2025-08-04
> > >
> > > Thank you very much for your thoughtful response. We're glad to hear your positive recognition of our technical contributions, and we will incorporate your general feedback into the final version.
> > >
> > > Regarding your follow-up questions, we would like to clarify that the paper’s contribution extends beyond developing a “data-driven sophisticated MNL that deals with IIA without using nested logit or other typical parametric models.”
> > >
> > > On the ML side:
> > >
> > > (1) Our model provides an efficient parameterization for finding the best k-order approximation of a universal logit for any k. This generalizes the work of Seshadri et al. [1], which focused on first-order (pairwise) interactions, despite mentioning higher-order decompositions in the work.
> > >
> > > (2) Interest in choice modeling within the ML community has been largely driven by practical applications—such as recommender systems, online marketplaces, and personalized decision-making [2–5]. These settings demand models capable of predicting user choices from item sets that are often complex, high-dimensional, and context-sensitive. While classical discrete choice models offer a principled foundation, they often lack the flexibility to address these real-world challenges. This motivated the development of machine learning-based choice models that retain interpretability while offering greater scalability and accuracy [6–9].
> > >
> > > More recently, the rise of large language models has renewed interest in choice modeling, highlighting the relevance of discrete choice theory for understanding and guiding complex decision-making processes [10]. While our work does not claim to directly solve these emerging challenges, we hope it helps bridge discrete choice theory and modern ML systems, and encourages further research in this direction.
> > >
> > > On the DCM side:
> > >
> > > As noted by Maragheh et al. [11], the nested logit model—while relaxing IIA—still cannot fully capture context effects. Among classical models, we focused our comparison on the Contextual MNL, which, to our knowledge, is the primary model explicitly designed to address context dependence. We also include comparisons with RUMnet [12], a neural representation of the Random Utility Model, which subsumes the nested and mixed logit models [13]. These choices reflect our intent to benchmark against strong baselines from both traditional and data-driven perspectives.
> > >
> > > We are fully aware of the discrete choice modeling community’s long-standing attention to context effects (as cited in our Introduction). Our work is not an attempt to “bypass reviewers from the core DCM community.” In fact, we intend to submit an extended version to a leading DCM journal, as we believe our contribution is meaningful and relevant to that audience.
> > >
> > > References:
> > >
> > > [1] Seshadri, A., Peysakhovich, A., & Ugander, J. (2020). *Discovering Context Effects from Raw Choice Data*. arXiv preprint arXiv:1902.03266.
> > > [2] John R. Birge, Xiaocheng Li & Chunlin Sun. *Learning from Stochastically Revealed Preference*. Advances in Neural Information Processing Systems 35 (NeurIPS 2022).
> > > [3] Yifan Feng & Yuxuan Tang. *On a Mallows-type Model for (Ranked) Choices*. Advances in Neural Information Processing Systems 35 (NeurIPS 2022).
> > > [4] Hyun-jun Choi, Rajan Udwani & Min-hwan Oh. *Cascading Contextual Assortment Bandits*. Advances in Neural Information Processing Systems 36 (NeurIPS 2023).
> > > [5] Recep Yusuf Bekci. *Online Learning of Delayed Choices*. Advances in Neural Information Processing Systems 37 (NeurIPS 2024).
> > > [6] Kiran Tomlinson and Austin R Benson. *Learning interpretable feature context effects in discrete choice*. In Proceedings of the 27th ACM SIGKDD Conference on Knowledge Discovery & Data Mining, pp. 1582–1592, 2021.
> > > [7] Nir Rosenfeld, Kojin Oshiba, and Yaron Singer. *Predicting choice with set-dependent aggregation*. In International Conference on Machine Learning, pp. 8220–8229. PMLR, 2020.
> > > [8] Amanda Bower and Laura Balzano. *Preference modeling with context-dependent salient features*. In International Conference on Machine Learning, pp. 1067–1077. PMLR, 2020.
> > > [9] Arjun Seshadri, Stephen Ragain, and Johan Ugander. *Learning rich rankings*. Advances in Neural Information Processing Systems, 33:9435–9446, 2020.
> > > [10] Jiang, R., Chen, K., Bai, X., He, Z., Li, J., Yang, M., Zhao, T., Nie, L., & Zhang, M. (2024). *A Survey on Human Preference Learning for Large Language Models*. arXiv preprint arXiv:2406.11191.
> > > [11] Reza Yousefi Maragheh, Xin Chen, James Davis, Jason Cho, Sushant Kumar, and Kannan Achan. *Choice modeling and assortment optimization in the presence of context effects*. Available at SSRN 3747354, 2020.
> > > [12] Ali Aouad and Antoine Désir. *Representing Random Utility Choice Models with Neural Networks*, July 2023. URL http://arxiv.org/abs/2207.12877.
> > > [13] McFadden, D. L., & Train, K. E. (2000). *Mixed MNL models for discrete response*. Journal of Applied Econometrics, 15(5), 447–470.

---

> > > > ### Comment · Reviewer_6ydQ · 2025-08-05
> > > >
> > > > Thanks again for this nice discussion. The fact that "RUMnet [12]" can (implicitly) "represent nested and mixed logit models [13]" to me does not imply that benchmarking against it is equivalent to a direct empirical comparison with nested logit or mixed logit themselves. These classical models have well-studied properties—particularly in terms of identifiability, robustness to specification, and interpretability—that are still unresolved in many NN approaches. Therefore, I still believe that including nested logit as a baseline would have strengthened the empirical framing of the paper, especially given its status as the canonical alternative for addressing IIA violations.
> > > >
> > > > I am also happy to hear that the authors also want to validate and publish this work in a classical DCM venue.

---

> > > > > ### Author Response · Authors · 2025-08-05
> > > > >
> > > > > Following your suggestion, we tested the nested logit model on the SFOshop and SFOwork datasets. As is well-known, specifying the nesting structure is a often a pratically challenging task. Fortunatelly, for this dataset, we are able to follow the guidelines provided in Koppelman and Bhat (2006). The results are presented below.
> > > > >
> > > > > We observe that the nested logit model is consistently outperformed by all context-dependent choice models, including our proposed DeepHalo model. Note also that its performance falls between that of MNL and mixed logit (not RUMNet but the original one), which is consistent with our intuition. This highlights the importance of capturing context effects when modeling the complex choice behavior observed in real-world datasets.
> > > > >
> > > > > Additionally, we plan to include a paragraph in the introduction to discuss the differences between traditional IIA-relaxing models and our approach. We hope this will help provide readers with a clearer and more intuitive understanding of our contribution.
> > > > >
> > > > > | Model             | SFOshop (Train/Test) | SFOwork (Train/Test) |
> > > > > |-------------------|---------------------|---------------------|
> > > > > | MNL               | 1.7281 / 1.7262     | 0.9423 / 0.9482     |
> > > > > | MLP               | 1.5556 / 1.5523     | 0.8074 / 0.8120     |
> > > > > | CMNL              | 1.5676 / 1.5686     | 0.8116 / 0.8164     |
> > > > > | Mixed MNL         | 1.5599 / 1.5577     | 0.8092 / 0.8153     |
> > > > > | Nested Logit      | 1.5788 / 1.5875     | 0.8366 / 0.8457     |
> > > > > | FateNet           | 1.5726 / 1.5765     | 0.8133 / 0.8167     |
> > > > > | TCNet             | 1.5694 / 1.5742     | 0.8114 / 0.8145     |
> > > > > | DeepHalo (Ours)   | 1.5385 / 1.5263     | 0.8040 / 0.8066     |
> > > > >
> > > > > Reference:
> > > > > Koppelman, F. S., & Bhat, C. (2006). *A self-instructing course in mode choice modeling: Multinomial and nested logit models.*

---

> > > > > > ### Comment · Reviewer_6ydQ · 2025-08-06
> > > > > >
> > > > > > Excellent. Congrats for a great paper. I will raise the score.

---

### Official Review · Reviewer_B9Eq · 2025-07-02

**Clarity:** 3
**Significance:** 2
**Originality:** 2
**Rating:** 5
**Confidence:** 4

**Summary:**

The authors propose a neural discrete choice model with context effects that handles item features. They use a Batsell-Polking style decomposition of context effects, but parameterize that decomposition with neural networks. The model applies iterative layers to capture higher-order context effects. By using a quadratic activations, the authors also provide an approach that captures higher-order context effects with only logarithmically many layers. In experiments on synthetic data, the authors demonstrate how their approach can recover planted similarity effects and learn very high-order interactions with low depth. On real datasets, they demonstrate better held-out performance than existing context effect models.

**Questions:**

1. Section 4.1 should explicitly say that (8) lets you capture $\ell$th order interactions with only $log_2 \ell$ layers, and should explain why.
2. Do you recommend using (8) in general? Are there settings where we should use (5) instead? Adding some guidance to the paper here would be helpful.
3. The intro says DeepHalo is more interpretable than the approaches of [Wong and Farooq, 2021], [Pfannschmidt et al., 2022], [Wang et al., 2023, Peng et al., 2024]. Similarly, the results state "while offering favorably improved interpretability compared to other deep neural network models." But using (10), can't we interpret context effects from any of the existing neural context effect models in the same way? What makes DeepHalo more interpretable?
4. The experiments of [Rosenfeld et al., 2020] use item features too---despite 1.1 saying they focus on the featureless case. What's the advantage of DeepHalo over SDA? DeepHalo seems like a Set-Dependent Embedding approach [Rosenfeld et al., 2020; eq. (2)], right?

Questions 3 and 4 are why I gave a 4 instead of a 5, and I'm very interested to hear the authors' response.

**Ethical Concerns:**

["NO or VERY MINOR ethics concerns only"]

**Final Justification:**

Thanks for the responses. I think this is a nice paper and I’d be happy to see it accepted. I’ve updated my score to a 5.

**Limitations:**

The discussion of limitations is limited. How does the sample complexity and need for hyperparameter tuning compare to existing approaches?

**Paper Formatting Concerns:**

I suggest replace $l$ with $\ell$ for $\ell$egibility.

**Quality:**

3

**Strengths And Weaknesses:**

Strengths:
1. Clearly written paper
2. Interesting approach, combining the Batsell-Polking approach with a neural network parametrization and an idea for efficiently capturing higher-order effects
3. Solid experiments

Weaknesses:
1. Could be clearer what the advantages are of the proposed method vs existing neural context effect models

Quality: good. Background and related work are thorough. Motivation and methods are good, baselines are pretty thorough and experiments are well done.

Clarity: There are a few points that can be clarified (see questions). Overall, though, the paper is clearly written. The title, though, is a bit generic, and could apply to several of the cited papers; I would suggest something that highlights the distinction of this approach vs those of Rosenfeld et al., Pfannschmidt et al, etc.

Significance: this is an important problem that has received significant attention, and the proposed method performs well in experiments.

Originality: reasonable, applies a different approach to neural modeling of context effects. I have some questions about what distinguishes DeepHalo from existing neural context effect models (see below)

Minor:
- Deep Sets double cite in Line 79

---

> ### Author Rebuttal · Authors · 2025-07-31
>
> We truly appreciate Reviewer B9Eq’s positive feedback. To address the questions and concerns about our method, we provide point-wise responses as follows.
>
> ### Weakness and Limitation
>
> > Q: Could be clearer what the advantages are of the proposed method vs existing neural context effect models
>
> A: The key advantage of DeepHalo lies in its ability to **explicitly control the maximum order of context (halo) effects**, enabling a flexible trade-off between model expressiveness and interpretability—something existing neural models lack.
>
> - First-order models (e.g., CMNL [Yousefi Maragheh et al., 2020], FETA [Pfannschmidt et al., 2022]) are interpretable but limited to pairwise interactions and cannot scale to higher-order effects due to computational constraints.
> - High-capacity models (e.g., FATE [Pfannschmidt et al., 2022], ResLogit [Wong and Farooq, 2021], and TCNet [Wang et al., 2023]) are expressive but entangle all item features, making it infeasible to control interaction order. Recovering meaningful low-order effects from these models requires evaluating up to $O(2^{|S|-1})$ terms.
>
> In contrast, DeepHalo models context effects in a **structured and decomposable** way. It allows practitioners to specify the maximum interaction order $K$, with a complexity of $O(2^K)$ to recover the effect terms, making it both **efficient and interpretable**. This is especially useful in real-world settings like recommendation and bundle pricing, where lower-order effects are often sufficient and desirable.
>
> > Q: The title, though, is a bit generic, and could apply to several of the cited papers; ...
>
> A: Thanks for your suggestion! We will consider adding some unique elements to our title that help distinguish it from other context-dependent model papers.
>
> > Q: How does the sample complexity and need for hyperparameter tuning compare to existing approaches?
>
> A: Regarding sample complexity and data efficiency, we refer the reviewer to our response to Reviewer Unm8 (Q: About data efficiency), where we provide an experiment evaluating model performance under varying data proportions.
>
> As for the hyperparameter tuning, we find empirically that DeepHalo is relatively easy to tune. **Once the maximum effect order is specified, the overall effect complexity is fixed**, which is the main tuning difference compared to existing approaches. Additional parameters are then introduced only to improve the approximation accuracy of these effects, rather than to increase model capacity arbitrarily.
>
> #### 1. Clarifying Section 4.1 — why Equation (8) needs only $\log_2l$ layers to model $l$‑th‑order interactions
>
> We agree that Section 4.1 should state this property explicitly. Currently, this is mentioned in Section 4.2, and the explanation is deferred in the appendix.
> In the revision, we will:
>
> 1) **Add a sentence to Section 4.1**
>    > “Because each Equation (8) layer doubles the maximum attainable interaction order, an *l*‑th‑order effect can be represented with at most ⌈log₂ l⌉ stacked layers.”
>
> 2) **Provide an intuitive explanation**
>    - A single layer (Equation (8)) captures up to second‑order effects.
>    - Stacking two layers composes these interactions, yielding up to 4‑th‑order effects, and so on.
>    - After *k* layers the highest order is $2^{k}$, giving the logarithmic bound.
>
>
> #### 2. Practical guidance on when to use Equation (5) vs Equation (8)
>
> We appreciate this insightful question. Generally, we recommend:
>
> - **Using Eq. (5) when training stability is a concern or when higher-order context effects are not considered.**
> - **Using Eq. (8) to reduce network depth when higher-order interactions are considered.**
> - Combining them for a tunable trade-off.
>
> Here we provide some details for further intuition. One of the strengths of the DeepHalo architecture is its structural flexibility: the activation function σ can be instantiated as any polynomial function. (5) and (8) can be used interchangeably within the same model. In practice, they can be freely mixed—for example, stacking layers with linear and quadratic activations can achieve effective interaction orders such as $((1+1+1)\times2+1)\times2$, offering nuanced control over expressivity.
>
> - Higher-degree activations (like Eq. (8)) allow *faster growth* in representational power with depth, efficiently capturing higher-order effects.
> - However, very high-degree activations in deep networks may introduce *training instability*, such as exploding gradients.
> - Empirically, we observe that for the same representable order, both variants achieve similar accuracy. Yet, first-order activations (Eq. (5)) tend to be more stable, especially on noisy or real-world datasets.
>
> This modularity makes DeepHalo a unified and controllable framework for context-dependent choice modeling, in both feature-rich and featureless settings. We will add a guidance paragraph in the revision to clarify these trade-offs.
>
> #### 3. Clarifying what makes DeepHalo more interpretable than prior neural context-effect models
> We appreciate the reviewer’s thoughtful question. While many prior models can interpret the effects using Eq. (10), this is under two premises:
> - The model is expressive enough to include all interaction orders up to $(|S|-1)$ instead of some specific orders (which first-order models like CMNL and FETA cannot achieve)
> - Existing expressive models like FATE and TCNet require a computational complexity of $O(2^{|S|})$ for recovering all halo effects in the model.
>
> In contrast, **our expressive DeepHalo model can capture up to a specific order and only needs $O(2^K)$ computational time to obtain $k$-order interactions**, which provide more interpretability. For example, modeling third-order effects over a catalog of 100 items without such control would require handling over 160,000 subsets, which is computationally prohibitive and interpretively infeasible.
>
> Besides, **the exact-interaction-order control inherited in our model is of practical interest.** As a concrete example, consider a choice set $\{A, B, C, D\}$. Then, estimating how the subset $\{A, B\}$ influences the utility of item $C$, the function must depend only on the features of $A$, $B$, and $C$, and must not involve irrelevant items like $D$. This subset-specific conditioning is critical for interpretability, but is not enforced in existing context-dependent ML models.
>
> The reason the existing models cannot capture exactly up to $k$-order interactions is that they attempt to learn context by directly entangling all item features. For instance, FATE combines global context with individual features but cannot restrict interaction depth. TCNet’s attention mechanism inherently mixes all items, effectively modeling full-order interactions by design. As a result, these models do not formulate context effects in a structured, decomposable way, making interpretation difficult.
>
> On the contrary, **our model achieves more interpretability with a modular design**:
> - Each layer introduces exactly one additional interaction order or $\times 2$ using quadratic activation.
> - The maximum effect order is bounded by the network depth $L$.
> - Residual and polynomial structures guarantee halo-decomposable utility functions. We will give a simple example later for better illustration.
>
>
> #### 4. Clarifying the distinction between DeepHalo and SDA [Rosenfeld et al., 2020]
>
> Thank you for the insightful observation. While DeepHalo shares some high-level similarities with set-dependent embedding (SDA) approaches such as [Rosenfeld et al., 2020, Eq. (2)], we would like to emphasize several key distinctions in both motivation and design, some are similar to Point 3.
>
> DeepHalo is fundamentally grounded in halo effect theory, which prioritizes **interpretable higher-order interactions**. While simply incorporating context information (e.g., through contextual functions like $\phi$) is straightforward, interpretability requires controlling the maximum order of interactions—a key principle not enforced by SDA.
>
> As the reviewer rightly noted, if Eq. (10) were applied to arbitrary context-based architectures (e.g., SDA), the number of halo terms would grow exponentially with the size of the offer set. This would result in a combinatorial explosion of interaction terms that are computationally hard and practically uninterpretable.
>
> In contrast:
>
> - SDA and related models aggregate item features in a fully entangled way.
>   - This makes it virtually impossible to recover the halo effect without enumerating all subsets—an exponentially large and ill-posed task.
> - DeepHalo, by design:
>   - Uses residual connections and polynomial activations to explicitly control and construct the interaction order.
>   - Ensures that utilities are halo-decomposable and that item features remain disentangled.
>   - Caps model expressiveness to low-order interactions, preserving interpretability and generalization.
>
> In short, while SDA enables generic set-dependent context modeling, DeepHalo provides a theoretically-grounded and architecturally-enforced approach to **interpretable, order-controlled context effects**, which is essential for practical discrete choice applications.
>
> For points 3 & 4, we refer the reviewer to Appendix A2.2 and A2.3, as well as the illustrative example in Question 2 of our response to reviewer Unm8, for a detailed explanation of how DeepHalo supports structured, decomposable modeling of context effects.

---

> > ### Comment · Reviewer_B9Eq · 2025-08-01
> >
> > Thanks for the answers! I'm still not convinced about the inefficiency of using Eq. (10) to interpret the context effects encoded in methods like SDA, FATE, and TCNet. Yes, to recover all context effects encoded by these models, we'd need to check all subsets. But say we only wanted to reproduce Figure 1 for SDA. In that test case, the source set $T$ has cardinality at most 2. Can't we just test the context effects encoded by SDA for each of the $O(2^K)$ (with $K = 2$) subsets of side 1 or 2 and get the same interpretation from SDA? The model might also encode different (and inconsistent) higher-order effects, but if we only care about these small cases, that doesn't seem like a problem.
> >
> > I suppose the counterargument is that these "lower-order" effects would not tell us anything about what would happen in any larger choice set. But if context effects in human preferences are actually low-order, maybe we'd hope SDA and other such methods capture that?

---

> > > ### Author Response · Authors · 2025-08-02
> > >
> > > Thank you very much for your follow-up question.
> > >
> > > Your observation is excellent and actually highlights one of the key benefits of our model. Indeed, for **small assortments**, one can evaluate subsets of size 1 or 2 to recover **low-order effects**. However, when facing **large assortment sizes**, the statement *“context effects in human preferences are actually low‑order”* is **not an assumption built into models like SDA**.
> > >
> > > DeepHalo explicitly enforces order control, which ensures that even in **large assortments**, the **low-order effects** can be learned **without contamination from uncontrolled high-order interactions**. In contrast, methods like SDA learn low-order effects implicitly, under the influence of many high-order interactions, which can introduce significant bias if you don't want to consider high-order effects.
> > >
> > > This problem becomes **especially critical when small-assortment cases (|S| = 2, 3) are rare in the dataset (as is often true in practice)**. In such scenarios, we must rely primarily on **large-assortment samples** to learn **low-order effects**, and without explicit order control, the learned effects will be heavily biased.

---

> > > ### Author Response · Authors · 2025-08-02
> > > **Further elaboration on the contribution**
> > >
> > > To help the reviewer better understand our **contribution**, we frame it as follows:
> > >
> > > Existing models are not designed to efficiently answer a common and practically important question:
> > > **What is the best K‑order approximation of a context‑dependent choice model, for arbitrary K ≥ 2?**
> > >
> > > This question frequently arises in e‑commerce applications, where practitioners often need **interpretable low‑order approximations** of complex choice behaviors.
> > >
> > > Moreover, the coefficients produced by models like **SDA** do not necessarily correspond to the best K‑order approximation. Simply **truncating** a higher‑order model does not guarantee that the resulting representation is the **optimal K‑order approximation**.

---

> > > > ### Comment · Reviewer_B9Eq · 2025-08-03
> > > >
> > > > Thanks for the clarification!

---

> > > > > ### Author Response · Authors · 2025-08-08
> > > > >
> > > > > Dear Reviewer,
> > > > >
> > > > > We would like to present an additional controlled experiment to illustrate the importance of using interpretable low-order approximations, as opposed to relying on post hoc truncation based on Equation (3).
> > > > >
> > > > > We generated a dataset of 10 items where the ground-truth utility function includes interaction effects up to the second order only. This setting is meant to mimic practical scenarios in which interaction effects are limited to low orders. We compared two modeling approaches for estimating first- and second-order effects:
> > > > >
> > > > > (1) DeepHalo: Fit our model with architecture explicitly constrained to capture up to second-order interactions. Report the in-sample negative log-likelihood (NLL) of the observed choices using the fitted model.
> > > > >
> > > > > (2) MLP with Post Hoc Truncation: Fit a standard MLP with the same number of learnable parameters. Use Equation (3) to extract interaction effects of all orders, then truncate post hoc to include only terms up to order $k$. Evaluate the in-sample NLL using only the truncated interaction effects.
> > > > >
> > > > > The in-sample NLL results for various truncation orders k are shown below:
> > > > >
> > > > >
> > > > > | Truncation Order $k$| MLP In-sample NLL | DeepHalo In-sample NLL |
> > > > > |----------------------------|---------------|---------------------|
> > > > > | 2                          | 0.7643        | 0.0922              |
> > > > > | 3                          | 1.6429        | 0.0922              |
> > > > > | 4                          | 0.8172        | 0.0922              |
> > > > > | 5                          | 0.6719        | 0.0922              |
> > > > > | 6                          | 0.3280        | 0.0922              |
> > > > > | 7                          | 0.1502        | 0.0922              |
> > > > > | 8                          | 0.0958        | 0.0922              |
> > > > > | 9                          | 0.0910        | 0.0922              |
> > > > >
> > > > > We observe that:
> > > > >
> > > > > - The MLP matches DeepHalo’s performance only when all interaction terms—including spurious higher-order ones—are included (i.e., no truncation).
> > > > >
> > > > > - Interestingly, increasing the truncation order beyond the true interaction order can worsen performance (e.g., from $k=2$ to $k=3$).
> > > > >
> > > > > This happens because the interaction effects of different orders operate jointly. When higher-order terms are learned by the model but not included in evaluation, they can distort the predictions. In contrast, DeepHalo’s explicit order control ensures a faithful low-order approximation without relying on fragile post hoc truncation.
> > > > >
> > > > > We hope this simple experiment clarifies the practical importance of interpretable, order-controllable models like DeepHalo. Thank you again for your thoughtful feedback.

---

> > > > > > ### Comment · Reviewer_B9Eq · 2025-08-08
> > > > > >
> > > > > > This is a very nice experiment that illustrates the point. I think this would be a great addition to the appendix, with a sentence or two discussing it in the paper.

---

> > > > > > > ### Author Response · Authors · 2025-08-09
> > > > > > >
> > > > > > > Thank you very much for the positive feedback! We will include this discussion in the paper as you suggested, along with a similar experiment based on a real-world dataset in our latest response to Reviewer Uum8.

---

### Official Review · Reviewer_Uum8 · 2025-07-03

**Clarity:** 3
**Significance:** 2
**Originality:** 1
**Rating:** 4
**Confidence:** 4

**Summary:**

Human revealed preferences are context-sensitive: Human choices are influenced by the context (made up of the set of alternatives) within which they are taken. These context effects cannot be explained by humans rationally choosing the alternative with the highest utility. One potential explanation for this phenomenon is that humans maximize utility, but that each alternatives’ utility is context dependent (dependent on the other items in the set of alternatives).

As this requires utility functions which maps arbitrary-size sets to real utility values, the learning problem becomes exponentially more complex. Prior work has attempted to solve this by decomposing the utility function, and has attempted to make computation cheaper by limiting calculations to lower-order interactions between alternatives. In line with this prior work, this paper proposes a similar decomposition of utility. It then proposes a new neural network architecture, DeepHalo, that can tractably calculates the decomposed utility up to arbitrarily high orders of interaction. This is done using a recursive formulation of context embeddings, where higher-order embeddings are produced from lower-order ones.

Experiments on synthetic data show the benefit of modeling higher-order effects with DeepHalo. On real choice data, the model’s predictions are shown to outperform baselines in terms of negative log-likelihood.

**Questions:**

Equation (3) includes the term $(-1)^{|T|-|R|}$, making each summand where the sizes of $R$ and $T$ differ by an uneven amount negative. I do not understand what this is supposed to achieve. What is the purpose of this?

In (3) you use $u_j(x_j, X_R)$. Here $u_j$ takes different arguments than in its previous uses. Is this the same function as defined in (2) (might I suggest writing it then as $u_j(X_{K \cup \\{j\\}})$, or is this a different function $u_j$?

For Tables 1 and 2, are the differences in test-set Negative log-likelihood (statistically) between DeepHalo and TCNet statistically significant? If I am reading Tables 5 and 6 in the appendix correctly then their mean NLLs are within 2 standard errors.

How sensitive is DeepHalo to the amount of training data? Would you say it is more of less data efficient than the other baselines?

**Ethical Concerns:**

["NO or VERY MINOR ethics concerns only"]

**Final Justification:**

The paper is technically solid but the contribution and novelty are fairly thin for NeurIPS. Nevertheless, this is compensated by the quality of the experiments and the value of being able to control interaction orders within the proposed method, pushing my judgement over the edge towards acceptance.

**Limitations:**

yes

**Quality:**

2

**Strengths And Weaknesses:**

The basic problem this paper tackles, modeling context effects, remains a very important problem in learning from human preferences. The amount of work that has been done in this area — and the paper does a very good job of covering the relevant literature here — shows how important it is. The writing of the paper is very clear, making it easy to understand the proposed model. The experiments are also excellent, covering a variety of datasets and comparing to a large set of baselines.

My main concern with this paper is novelty. As far as I can see, the specific decomposition of the utility function which DeepHalo uses mirrors FETA’s almost exactly. The proposed neural network architecture is a fairly straightforward variation on DeepSets. The authors claim that the model offers an increased level of interpretability. However, the notion of relative context effect in (10) used to interpret the model can be calculated with any model (including prior ones) that decomposes the utility of a choice set into contributions of subsets; the interpretability is therefore not derived from the DeepHalo architecture specifically.

Furthermore, I’m not convinced that the proposed network architecture replicates the utility decomposition the paper aims for (line 189). It seems correct for pairwise interactions, but I do not see how the recursive structure proposed replicates this utility decomposition.

Minor points:
- Line 79: duplicate instance of “Deep Sets [Zaheer et al., 2018]
- Lines 180-182. Misspelling of FETA and FATE. In the original paper the “E” stands for “evaluate” not “evaluation”.
- At various points in the paper, the authors use $S$ both to indicate the set $S$ and the size of that set. This is needlessly confusing. It would be better to use $|S|$ for the size of $S$ consistently.

---

> ### Author Rebuttal · Authors · 2025-07-31
>
> We sincerely appreciate your recognition of our problem modeling, methodological soundness, real-world validation, and thoughtful baseline selection.
>
> To address the concerns, we provide detailed point-wise responses below.
>
>
> > **Q: As far as I can see, the specific decomposition of the utility function which DeepHalo uses mirrors FETA’s almost exactly...**
>
> A: The key methodological contribution of DeepHalo lies in introducing a **structured and order-controllable framework** that balances expressiveness and interpretability—a capability that is lacking in existing context-dependent choice models. Among context-dependent models,
>
> - First-order models such as CMNL (Yousefi Maragheh et al., 2020), low-rank Halo MNL (Ko and Li, 2024), and FETA (Pfannschmidt et al., 2022) are **inherently restricted to pairwise interactions**. For instance, FETA proposes a decomposition that resembles the halo effect, but in practice, it is limited to first-order pairwise interactions. The architecture is not scalable to higher-order effects, as doing so would require exponentially large tensor storage and computational resources, rendering it impractical for real-world applications.
>
> - Other higher-order models, such as FATE (Pfannschmidt et al.,2022) and TCNet (Wang et al., 2023), attempt to learn context effects by **directly entangling all item features** within the offer set. While their outputs can be retrospectively interpreted as halo effects via Eq. (10), the interaction order is not explicitly controlled. Recovering a meaningful decomposition in these models would require computational complexity on the order of $O(2^{∣S∣−1})$, due to the exponential number of possible interaction terms. This limitation stems from their architectural design: context effects are not modeled in a structured or decomposable way. Specifically, FATE aggregates all item features into a set embedding, which is then combined with individual item features through complex nonlinear transformations, obscuring the origin and order of interactions. TCNet, similarly, uses attention mechanisms where the softmax denominator implicitly mixes information from all items in the offer set, making it difficult to isolate individual effects. As a result, neither model can recover a truly decomposable utility function.
>
> In contrast, DeepHalo is explicitly designed to address this gap by enabling **precise control over the maximum order of halo interactions**. To the best of our knowledge, it is the first model that supports flexible trade-offs between model complexity and interpretability in this context.
>
> In e-commerce applications such as recommendation systems and bundle pricing, practitioners are often primarily interested in lower-order context effects, which are more stable, interpretable, and practically useful. While expressive models can theoretically recover such effects, they do so implicitly as part of an entangled representation involving many higher-order interactions. Consequently, low-order effects cannot be meaningfully isolated. DeepHalo offers a principled approximation that explicitly models limited-order halo effects. It allows users to control the maximum interaction order $K$, with computational complexity only $O(2^K)$, offering a **flexible trade-off between efficiency and expressiveness**.
>
> Our contribution goes beyond what Deep Set offers. While DeepSets provides permutation invariance, DeepHalo integrates domain-specific insights from discrete choice theory and the halo literature, along with architectural elements like residual connections and polynomial activations. This enables interpretable, high-order interaction modeling in a principled and controllable way.
>
>
> > **Q: Furthermore, I’m not convinced that the proposed network architecture replicates the utility decomposition the paper aims for (line 189)...**
>
> A: Thank you for raising this point. The key intuition is that every recursion depth l in Eqs. (4)–(5) adds **exactly one additional order of interaction** while preserving all lower‑order terms via the residual connection. The process is the same as Appendix A 2.2 and A 2.3. To better facilitate the understanding, we provide a simple example below.
>
> Consider an offer set ${j, k, l}$ and let $\phi$ be an identity mapping. We focus on the change of $z_{j}$ in DeepHalo.
> 1. **Pairwise (1‑st order) layer.**
>    The first layer aggregates the linear summary vector $\bar Z^{1}$ from raw embedding vector $z^{0}$ and modulates them with $z_{j}^{0}$. This yields
>    $z_{j}^{1}=z_{j}^{0} + \frac1H\sum_{h}\bar Z_{h}^{1} \cdot z_{j}^{0}$, which contains only **pairwise interactions** between the target alternative $j$ and every other item $k, l$ in the set.
>    If we denote $f_j$ as any function containing only item j's feature information, the above output can be formulated as,
>    $z_{j}^{1} = f_j + \frac1H\sum_{h}(f_j + f_k + f_l) \cdot f_j$, thus it contains zero order term $f_j$ and first order term $f_k \cdot f_j$ and $f_l \cdot f_j$, second order term $f_j \cdot f_k \cdot f_l$ won't appear. For simplicity, we use $f_{ab}$ to represent any term like $f_a \cdot f_b$ and $f_b \cdot f_a$, which only contains the features of item a and item b.
>
> 2. **Second‑order layer.**
>    After the first layer, $z_{k}^{1}$ and $z_{l}^{1}$ already embed their pairwise effects with every other item.
>    The second layer builds
>
>    $\bar Z^{2}=\tfrac1S\sum_{m}W^{2}z_{m}^{1}$, which contains term $f_j, f_k, f_l, f_{jk}, f_{jl}, f_{kl}$.
>
>    $z_{j}^{2}=z_{j}^{1}+\tfrac1H\sum_{h} \bar Z_{h}^{2} \cdot z_{j}^{0}$ which is analogous to $(f_j + f_{jk} + f_{jl}) + \sum_h(f_j + f_k + f_l + f_{jk} + f_{jl} + f_{kl}) \cdot f_j$,
>    which brings about second order term $f_{jkl} = f_{kl} \cdot f_{j}$. It can be interpreted as the effect of ${k,l}$ on $j$.
>
> This example illustrates how we can add up orders by increasing the number of layers and get a decomposable utility representation. If we change the $\cdot$ in layers to some other polynomial function, the term order will grow faster, like in the first layer we change to $(f_j + f_k + f_l)^2 \cdot f_j$, in the sum will directly bring about the second order term $f_{jkl}$. The quadratic activation naturally brings a $log_2$ form depth requirement.
>
> > **Q: The purpose of $(-1)^{|T|-|R|}$.**
>
> A: The $(−1)^{∣T∣−∣R∣}$ factor in Eq. (3) is the **inclusion–exclusion inversion coefficient** on the Boolean lattice of subsets. Our $v_j(⋅)$ must invert the cumulative‑sum relation $u_j(X_S)=\sum_{T\subset S\setminus\{j\}} v_j(X_{T\cup\{j\}})$; effects contributed to many supersets are otherwise over‑counted. Alternating signs guarantee that terms appearing an even number of times cancel and those appearing an odd number of times net to the correct single contribution, yielding a unique decomposition.
>
> > **Q: About Performance Difference between DeepHalo and TCnet...**
>
> A: Thanks for the insightful observation. We acknowledge that similar performance between DeepHalo and TCNet is expected; however, we wish to emphasize a key difference in their modeling philosophies.
>
> While TCNet can approximate utility decompositions (maximum order), its Transformer architecture lacks control over the maximum interaction order. Attention layers aggregate across the full set, leading to large quantities of high-order effects that are difficult to interpret. In contrast, DeepHalo explicitly controls interaction order through its recursive structure, enabling better interpretability and identifiability.
>
> Given that low-order interactions often dominate in real-world settings, it is not surprising that TCNet matches DeepHalo's accuracy. However, DeepHalo achieves this with controlled expressiveness, offering a more interpretable alternative. Rather than outperforming black-box models in accuracy, our main goal is to provide a principled, order-aware design that balances predictive performance with transparency.
>
> > **Q: About data efficiency**
>
> | Ratio (%) | CMNL  | MNL   | FATE   | MLP   | MixedMNL | TCnet | **DeepHalo** |
> |-----------|-------|-------|--------|-------|-----------|--------|--------------|
> | 10        | 1.574 | 1.898 | 1.577 | 1.571 | 1.674     | 1.589 | **1.566**     |
> | 20        | 1.567 | 1.869 | 1.576 | 1.563 | 1.617     | 1.573 | **1.558**     |
> | 30        | 1.564 | 1.844 | 1.574 | 1.557 | 1.590     | 1.569 | **1.551**     |
> | 40        | 1.562 | 1.820 | 1.574 | 1.554 | 1.576     | 1.568 | **1.548**     |
> | 50        | 1.561 | 1.779 | 1.576 | 1.550 | 1.566     | 1.570 | **1.544**     |
> | 60        | 1.562 | 1.760 | 1.576 | 1.545 | 1.563     | 1.551 | **1.537**     |
> | 70        | 1.555 | 1.742 | 1.576 | 1.542 | 1.560     | 1.539 | **1.532**     |
> | 80        | 1.544 | 1.726 | 1.576 | 1.537 | 1.558     | 1.560 | **1.528**     |
> | 90        | 1.550 | 1.699 | 1.576 | 1.539 | 1.555     | 1.549 | **1.526**     |
> | 100       | 1.550 | 1.726 | 1.560 | 1.537 | 1.558     | 1.538 | **1.528**     |
>
> To evaluate data efficiency, we subsample the SFOshop dataset—known for strong context effects—at ratios from 10% to 100%, training all neural models under a strict ~1k parameter budget.
>
> Across all settings, **DeepHalo consistently achieves the lowest Test NLL**, showing strong generalization even with limited data. Its Test NLL steadily improves with more data, while other models (TCNet, MLP, FATE) either plateau early or fluctuate.
>
>
> ### **Clarification**
> The expression $u_j(x_j, X_R)$ in Eq. (3) refers to the same function as $u_j(X_{R \cup \{j\}})$ in Eq. (2), and we will revise the notation for consistency. We will also correct the duplicate citation of Deep Sets [Zaheer et al., 2018], fix the misspelling of FETA and FATE (“E” stands for “evaluate”), and use $|S|$ consistently to denote set size. These issues will be addressed in the final version. We appreciate the reviewer’s careful reading.

---

> > ### Comment · Reviewer_Uum8 · 2025-08-04
> >
> > Dear authors,
> >
> > Thank you for the detailed responses to my questions and remarks. Thanks to your responses I now understand the motivation for the proposed architecture in terms of controlling the order of interactions, and I'm confident that the paper is methodologically solid.
> >
> > I think this is an excellent paper. However - and I agree with reviewer 6ydQ here - within the wider NeurIPS context this paper's contribution over prior work is quite incremental and its impact is limited to the discrete choice modeling niche. If I evaluate the paper purely within this discrete choice modeling context, I think the novelty over FATE is sufficient, but the predictive improvement is marginal. I understand the argument that for practitioners using these models there are advantages from being able to control the order of interactions, but I would have expected more significant experimental evidence to support this claim.
> >
> > I have raised my score, but for the reasons above I cannot quite recommend acceptance.

---

> > > ### Author Response · Authors · 2025-08-08
> > >
> > > Dear Reviewer,
> > >
> > > To provide further experimental evidence supporting the benefit of controlling interaction order, we conducted the following controlled experiment.
> > >
> > > We generated a dataset of 10 items where the ground-truth utility function includes interaction effects up to the second order only. This setting is meant to mimic practical scenarios in which interaction effects are limited to low orders. We compared two modeling approaches for estimating first- and second-order effects:
> > >
> > > (1) DeepHalo: Fit our model with architecture explicitly constrained to capture up to second-order interactions. Report the in-sample negative log-likelihood (NLL) of the observed choices using the fitted model.
> > >
> > > (2) MLP with Post Hoc Truncation: Fit a standard MLP with the same number of learnable parameters. Use Equation (3) to extract interaction effects of all orders, then truncate post hoc to include only terms up to order $k$. Evaluate the in-sample NLL using only the truncated interaction effects.
> > >
> > > The in-sample NLL results for various truncation orders k are shown below:
> > >
> > >
> > > | Truncation Order $k$| MLP In-sample NLL | DeepHalo In-sample NLL |
> > > |----------------------------|---------------|---------------------|
> > > | 2                          | 0.7643        | 0.0922              |
> > > | 3                          | 1.6429        | 0.0922              |
> > > | 4                          | 0.8172        | 0.0922              |
> > > | 5                          | 0.6719        | 0.0922              |
> > > | 6                          | 0.3280        | 0.0922              |
> > > | 7                          | 0.1502        | 0.0922              |
> > > | 8                          | 0.0958        | 0.0922              |
> > > | 9                          | 0.0910        | 0.0922              |
> > >
> > > We observe that:
> > >
> > > - The MLP matches DeepHalo’s performance only when all interaction terms—including spurious higher-order ones—are included (i.e., no truncation).
> > >
> > > - Interestingly, increasing the truncation order beyond the true interaction order can worsen performance (e.g., from $k=2$ to $k=3$).
> > >
> > > This happens because the interaction effects of different orders operate jointly. When higher-order terms are learned by the model but not included in evaluation, they can distort the predictions. In contrast, DeepHalo’s explicit order control ensures a faithful low-order approximation without relying on fragile post hoc truncation.
> > >
> > > We hope this simple experiment clarifies the practical importance of interpretable, order-controllable models like DeepHalo. Thank you again for your thoughtful feedback.

---

> > > > ### Comment · Reviewer_Uum8 · 2025-08-08
> > > >
> > > > Dear Authors,
> > > >
> > > > Thank you for your continued efforts to improve this paper. I appreciate the additional experiments you have produced during this rebuttal phase, especially the comparison to nested logit models and these new results trying to show the value of the controllabiltiy of the interaction order in DeepHalo. I think the above experiment would have been even more interesting if it had been performed on human data, because the relevance of these results is now predicated on the assumption that within human decisions the interactions are limited to lower orders. Nevertheless, these results help me to appreciate this controllabiltiy within DeepHalo more than I had before.
> > > >
> > > > Based on the above and your continued discussions with the other reviewers, I will raise my score to borderline accept.

---

> > > > > ### Author Response · Authors · 2025-08-09
> > > > >
> > > > > We sincerely thank you for recognizing the additional experiments and for the insightful discussion.
> > > > >
> > > > > Regarding your suggestion — *“I think the above experiment would have been even more interesting if it had been performed on human data, because the relevance of these results is now predicated on the assumption that within human decisions the interactions are limited to lower orders”* — we fully agree. At the end of this discussion, we present a simple experiment on the SFOshop dataset to address this point.
> > > > >
> > > > > First, we tested the assumption that human decisions the interactions are limited to lower orders.
> > > > > We fitted a 2nd-order DeepHalo and a full-order DeepHalo (fitting only, without considering out-of-sample performance).  Their in-sample NLL losses were 1.5339 and 1.5331, respectively — essentially identical.
> > > > > This indicates that, for this real dataset, second-order effects are sufficient to capture human choice behavior.
> > > > >
> > > > > We then repeated the same truncation experiment as before and obtained similar outcomes:
> > > > >
> > > > > | Truncation Order \(k\) | MLP In-sample NLL | DeepHalo(2-order) In-sample NLL |
> > > > > |------------------------|-------------------|----------------------------------|
> > > > > | 2 | 1.6065 | 1.5339 |
> > > > > | 3 | 1.6168 | 1.5339 |
> > > > > | 4 | 1.6834 | 1.5339 |
> > > > > | 5 | 1.5806 | 1.5339 |
> > > > > | 6 | 1.5406 | 1.5339 |
> > > > > | 7 | 1.5340 | 1.5339 |

---

### Note · Authors · 2025-08-13

**Final Remarks**

We greatly appreciate the time, effort, and constructive feedback from all reviewers and the area chair. Here, we provide a concise summary of our contributions and the outcomes from the discussion phase.

**Contributions**
1. Proposed an order-controllable context-dependent choice model that balances interpretability and performance.
2. Extended the Halo effect framework to settings with item features.
3. Developed an identifiable effect recovery algorithm.

**Outcomes from the Discussion**

**1. Further clarification of novelty**

We further clarify the novelty of DeepHalo from three perspectives:

(i) Mechanism for controlling context effect order

(ii) Necessity of order control

(iii) Limitations of existing models

We explained (i) with a simplified example in our discussion with Uum8 and the derivation in the Appendix. Experiments on synthetic and real datasets—well recognized by reviewers Uum8 and B9Eq—further validated (i) (ii) (iii) and showed that:

- Lower- and higher-order effects work in tandem; uncontrolled order leads to exponentially more effects and reduced interpretability.
- Post-hoc truncation to obtain lower-order effects can severely distort predicted choice probabilities.
- Existing models, by entangling all item features, can only recover a full-order model, whereas our method enables faithful lower-order approximations.


**2. More baseline models, training details, and additional experiments**

We further improved the empirical evaluation by:
- Incorporating reviewer 6ydQ’s suggestion to include classical non-IIA models (mixed logit, nested logit) and report performance.
- Following reviewer D2h9’s advice to provide parameter counts and add baselines in featureless experiments.
- Responding to Uum8’s request by adding a data efficiency experiment.

Feedback from reviewers indicates these concerns have been satisfactorily addressed.

**3. Clarifications on presentation-related points**

Reviewers raised several minor points regarding presentation and confusing points. We have addressed these by:
- Providing point-by-point clarifications during the discussion.
- Outlining specific revision plans to improve clarity and readability.

**Summary**

Across the discussion phase, all reviewers reached consensus in giving positive evaluations of our work. Nearly all concerns raised were fully resolved through additional experiments, clarifications, and revision plans.

---

### Decision · Program_Chairs · 2025-09-17

**Decision:**

Accept (spotlight)

**Comment:**

This paper proposes a new method for context-dependent choice prediction the the HALO formulation of behavioral context effects. The paper is clearly written and the proposed method is interesting and novel. There was a long and fruitful discussion between the reviewers and the authors, which was helpful in addressing many of the reviewer's concerns, and shed light on the aspects of the paper that would likely be of interest to the NeurIPS community. The authors are highly encouraged to incorporate these conclusions and their new experiments into the final version of the paper.